# Assessing healthy vaccinee effect in COVID-19 vaccine effectiveness studies: a national cohort study in Qatar

Hiam Chemaitelly[1,2,3]*, Houssein H Ayoub[4], Peter Coyle[5,6,7], Patrick Tang[8], Mohammad R Hasan[9], Hadi M Yassine[5,10], Asmaa A Al Thani[5,10], Zaina Al-Kanaani[6], Einas Al-Kuwari[6], Andrew Jeremijenko[6], Anvar Hassan Kaleeckal[6], Ali Nizar Latif[6], Riyazuddin Mohammad Shaik[6], Hanan F Abdul-Rahim[11], Gheyath K Nasrallah[5,10], Mohamed Ghaith Al-Kuwari[12], Hamad Eid Al-Romaihi[13], Mohamed H Al-Thani[13], Abdullatif Al-Khal[6], Roberto Bertollini[12], Adeel A Butt[3,6,14], Laith J Abu-Raddad[2,3,11,15]*

[1]Infectious Disease Epidemiology Group, Weill Cornell Medicine-Qatar, Cornell University, Doha, Qatar; [2]World Health Organization Collaborating Centre for Disease Epidemiology Analytics on HIV/AIDS, Sexually Transmitted Infections, and Viral Hepatitis, Weill Cornell Medicine-Qatar, Cornell University, Qatar Foundation – Education City, Doha, Qatar; [3]Department of Population Health Sciences, Weill Cornell Medicine, Cornell University, New York, United States; [4]Mathematics Program, Department of Mathematics and Statistics, College of Arts and Sciences, Qatar University, Doha, Qatar; [5]Department of Biomedical Science, College of Health Sciences, QU Health, Qatar University, Doha, Qatar; [6]Hamad Medical Corporation, Doha, Qatar; [7]Wellcome-Wolfson Institute for Experimental Medicine, Queens University, Belfast, United Kingdom; [8]Department of Pathology, Sidra Medicine, Doha, Qatar; [9]Department of Pathology and Molecular Medicine, McMaster University, Hamilton, Canada; [10]Biomedical Research Center, QU Health, Qatar University, Doha, Qatar; [11]Department of Public Health, College of Health Sciences, QU Health, Qatar University, Doha, Qatar; [12]Primary Health Care Corporation, Doha, Qatar; [13]Ministry of Public Health, Doha, Qatar; [14]Department of Medicine, Weill Cornell Medicine, Cornell University, New York, United States; [15]College of Health and Life Sciences, Hamad bin Khalifa University, Doha, Qatar

*For correspondence:
hsc2001@qatar-med.cornell.edu (HC);
lja2002@qatar-med.cornell.edu (LJA-R)

## Abstract

**Background:** This study investigated the presence of the healthy vaccinee effect—the imbalance in health status between vaccinated and unvaccinated individuals—in two rigorously conducted COVID-19 vaccine effectiveness studies involving primary series and booster vaccinations. It also examined the temporal patterns and variability of this effect across different subpopulations by analyzing the association between COVID-19 vaccination and non-COVID-19 mortality in Qatar.

**Methods:** Two matched, retrospective cohort studies assessed the incidence of non-COVID-19 death in national cohorts of individuals with a primary series vaccination versus no vaccination (two-dose analysis), and individuals with three-dose (booster) vaccination versus primary series vaccination (three-dose analysis), from January 5, 2021, to April 9, 2024.

**Results:** The adjusted hazard ratio (aHR) for non-COVID-19 death was 0.76 (95% CI: 0.64–0.90) in the two-dose analysis and 0.85 (95% CI: 0.67–1.07) in the three-dose analysis. In the first 6 months of follow-up in the two-dose analysis, the aHR was 0.35 (95% CI: 0.27–0.46); however, the

combined analysis of all subsequent periods showed an aHR of 1.52 (95% CI: 1.19–1.94). In the first 6 months of follow-up in the three-dose analysis, the aHR was 0.31 (95% CI: 0.20–0.50); however, the combined analysis of all subsequent periods showed an aHR of 1.37 (95% CI: 1.02–1.85). The overall effectiveness of the primary series and third-dose vaccinations against severe, critical, or fatal COVID-19 was 95.9% (95% CI: 94.0–97.1) and 34.1% (95% CI: –46.4–76.7), respectively. Subgroup analyses showed that the healthy vaccinee effect is pronounced among those aged 50 years and older and among those more clinically vulnerable to severe COVID-19.

**Conclusions:** A pronounced healthy vaccinee effect was observed during the first 6 months following vaccination, despite meticulous cohort matching. This effect may have stemmed from a lower likelihood of vaccination among seriously ill, end-of-life individuals, and less mobile elderly populations.

**Funding:** Biomedical Research Program and the Biostatistics, Epidemiology, and Biomathematics Research Core, and Junior Faculty Transition to Independence Program, all at Weill Cornell Medicine-Qatar, Qatar University, Ministry of Public Health, Hamad Medical Corporation, Sidra Medicine, Qatar Genome Programme, Qatar University Biomedical Research Center, and L'Oréal-UNESCO For Women In Science Middle East Regional Young Talents Program.

## Editor's evaluation

This fundamental work devoted to the effectiveness of COVID-19 vaccination is the first to calculate within a single paper the COVID vaccine effectiveness as well as a crucial confounder – the so-called healthy vaccinee effect/bias that influences results of observational vaccine effectiveness studies. Using rigorous methods and providing compelling evidence, the authors found a 65 % decrease in the likelihood of dying from non-COVID causes in the vaccinated individuals in the first six months after vaccination compared to the meticulously matched unvaccinated individuals. This indicates that observational studies on COVID-19 vaccines may inflate vaccine effectiveness, even if it is evaluated using the best available industry-standard methods. The work will be of broad interest not only to epidemiologists and vaccinologists but virtually to any scientist investigating the role of vaccines.

## Introduction

While randomized controlled trials (RCTs) remain the gold standard for determining vaccine efficacy, they often have short follow-up durations, primarily involve healthy participants, and may cover only a limited range of clinical outcomes (*Janiaud et al., 2021*; *Nelson et al., 2009*; *Simonsen et al., 2007*). Real-world observational studies are frequently utilized to evaluate vaccine effectiveness beyond the controlled trial environment (*Chambers, 2021*; *de Waure et al., 2024*; *Nelson et al., 2009*). In these settings, diverse health statuses, variable health behaviors, and structural determinants can influence vaccine uptake, potentially biasing estimates of effectiveness (*Chambers, 2021*; *de Waure et al., 2024*; *Nelson et al., 2009*).

The accuracy of vaccine effectiveness estimates from observational studies can be affected by bias arising from two opposing effects: the indication effect and the healthy vaccinee effect (*Remschmidt et al., 2015*; *Nelson et al., 2009*). The indication effect occurs when individuals with underlying health conditions are more likely to receive vaccination, potentially leading to an underestimation of vaccine effectiveness (*Remschmidt et al., 2015*). Conversely, the healthy vaccinee effect occurs when healthier or health-conscious individuals are more likely to receive vaccination, leading to an overestimation of vaccine effectiveness (*Remschmidt et al., 2015*). Both effects can bias and skew effectiveness results by conflating health status with the protective effects of the vaccine (*Remschmidt et al., 2015*; *Nelson et al., 2009*).

Such effects have been documented in observational studies of influenza vaccine effectiveness (*Remschmidt et al., 2015*; *Simonsen et al., 2005*; *Nelson et al., 2009*; *Jackson et al., 2006a*), coronavirus disease 2019 (COVID-19) vaccine effectiveness (*Fürst et al., 2024*; *Høeg et al., 2023*; *Xu et al., 2023*), and prescriptive medication effectiveness (*Nelson et al., 2009*). In particular, studies have documented a strong healthy vaccinee effect in estimations of influenza vaccine effectiveness among the elderly (*Jackson et al., 2006a*; *Nelson et al., 2009*). While it is common for studies to control for such effect by adjusting for coexisting conditions based on administrative healthcare utilization

**eLife digest** Before new vaccines are made widely available, their efficacy and safety are tested in laboratories and clinical trials. Once approved, researchers can continue to monitor how these vaccines perform in the 'real world' by analysing healthcare data. This can provide further insights over a longer timeframe and across a broader range of people.

However, these real-world analyses can be skewed by a range of factors. For example, if healthier people are more likely to receive the vaccine, researchers may overestimate its effectiveness due to fewer deaths or severe illnesses amongst that group. This 'healthy vaccinee effect' has been observed in influenza vaccines among older people, for example.

To determine whether the healthy vaccinee effect influenced COVID-19 vaccine studies, Chemaitelly et al. analysed national health records in Qatar. Groups of vaccinated and unvaccinated people were selected to have matching demographics, such as age, sex and number of preexisting conditions. Overall, the study confirmed strong protection from vaccination against severe forms of COVID-19.

However, the results showed that, compared to their unvaccinated peers with similar characteristics, vaccinated people were 65 per cent less likely to die due to reasons unrelated to COVID-19 in the next six months after having received the vaccination. This effect was even stronger among individuals aged 50 years or older, as well as those with clinical vulnerabilities. Chemaitelly et al. suggest that this likely reflects lower vaccine uptake among seriously ill, end-of-life individuals and less mobile older populations with short life expectancy. Future work may be needed to understand if this can be generalised to countries that differ from Qatar, whose population mainly consists of healthy adult migrant workers.

databases, this approach may not sufficiently or properly adjust for this effect, as it may not capture the illness severity, recency, duration, or the functional status of individuals—factors that can confound the association between vaccination and health outcomes, particularly in the elderly (*Jackson et al., 2006a*; *Remschmidt et al., 2015*; *Nelson et al., 2009*). Assessing coexisting conditions based on database variables can also be affected by differential misclassification, as these variables not only reflect chronic diseases but also inherently measure utilization of health services (*Nelson et al., 2009*).

A notable feature of the healthy vaccinee effect is its potentially strong time dependence, most pronounced immediately after vaccination but gradually diminishing (*Jackson et al., 2006a*; *Nelson et al., 2009*). This trend is observed because seriously ill individuals, those with deteriorating health, and frail, less mobile elderly persons are less likely to be vaccinated, resulting in a higher short-term mortality risk among the unvaccinated (*Jackson et al., 2006a*; *Nelson et al., 2009*). Studies show that elderly individuals who are more mobile or have fewer functional limitations are more likely to be vaccinated (*Nelson et al., 2009*; *Jackson et al., 2006b*). For instance, one study demonstrated that the inability to bathe independently was associated with a 13-fold increase in mortality risk and a 52% reduced likelihood of receiving a vaccination (*Jackson et al., 2006b*). Over time, as the less functional and seriously ill individuals in the unvaccinated group die, the disparities between the vaccinated and unvaccinated groups diminish (*Jackson et al., 2006a*; *Nelson et al., 2009*). Additional changes in health status over time among members of both groups also contribute to this equilibration (*Jackson et al., 2006a*; *Nelson et al., 2009*).

In this national retrospective cohort study, the presence of the indication or healthy vaccinee effects was investigated within a conventionally designed and well-controlled COVID-19 vaccine effectiveness study, covering both primary series and booster mRNA vaccinations. Three aspects of these effects were explored: their existence, their temporal pattern, and their variability across different subpopulations. This was achieved by assessing the association between COVID-19 vaccination and non-COVID-19 mortality, which serves as a suitable control outcome to gauge the degree of potential residual effect in well-controlled vaccine effectiveness estimates against SARS-CoV-2 infection or severe forms of COVID-19 (*Nelson et al., 2009*).

## Methods

### Study population and data sources

This study was conducted among the resident population of Qatar from January 5, 2021, which marks the earliest record of a completed COVID-19 primary series vaccination, to April 9, 2024, the study's end date. Data on COVID-19 laboratory testing, vaccination, hospitalization, and death were retrieved from the integrated, nationwide digital health information platform (Section S1 in Supplementary Appendix). Deaths not related to COVID-19 were sourced from the national federated mortality database, which captures all deaths in the country, occurring in healthcare facilities and elsewhere, including forensic deaths investigated by Qatar's Ministry of Interior.

The national digital health information platform includes all SARS-CoV-2-related records, encompassing COVID-19 vaccinations, hospitalizations, and polymerase chain reaction (PCR) tests, irrespective of location or facility, and, from January 5, 2022, medically supervised rapid antigen tests (Section S2). Until October 31, 2022, Qatar maintained an extensive testing approach, testing 5% of the population weekly, primarily for routine purposes such as screening or travel-related requirements (*Chemaitelly et al., 2021b*; *Altarawneh et al., 2022b*). From November 1, 2022, onward, testing was reduced to below 1% of the population weekly (*Chemaitelly et al., 2023a*). Most COVID-19 infections in Qatar were identified through routine testing rather than symptomatic presentation (Section S1) (*Altarawneh et al., 2022b*; *Chemaitelly et al., 2021b*). The national platform further contains data on coexisting conditions for individuals who have accessed care through the universal public healthcare system since the establishment of the digital health platform in 2013 (Section S3).

Demographic information was obtained from the national health registry. Qatar's demographic composition is distinct, with only 9% of the population aged 50 years or older and 89% being resident expatriates from over 150 countries (*Abu-Raddad et al., 2021a*). Further details on Qatar's population and COVID-19 databases have been previously published (*Chemaitelly et al., 2021b*; *Altarawneh et al., 2022b*; *Chemaitelly et al., 2023e*; *Abu-Raddad et al., 2021a*; *AlNuaimi et al., 2023*; *Chemaitelly et al., 2021a*).

### COVID-19 vaccination

COVID-19 vaccination in Qatar was predominantly conducted using mRNA vaccines and adhered to United States Food and Drug Administration-approved protocols throughout the pandemic (*Chemaitelly et al., 2021b*; *Altarawneh et al., 2022b*). Vaccines were provided free of charge to all individuals, regardless of citizenship status, exclusively through the public healthcare system (*Chemaitelly et al., 2021b*; *Altarawneh et al., 2022b*). The immunization campaign was launched on December 21, 2020, with the BNT162b2 vaccine (*Polack et al., 2020*), and 3 months later, the mRNA-1273 vaccine (*Baden et al., 2021*) was added. Most primary series vaccinations were administered in 2021 due to the rapid scale-up of mass vaccination efforts (*Figure 1—figure supplement 1A*).

The vaccine rollout was implemented in phases, prioritizing frontline healthcare workers, individuals with severe or multiple chronic conditions, and those aged ≥70 years (*Chemaitelly et al., 2021b*). Vaccination was subsequently extended to select professional groups, such as teachers, and then to the general population, beginning with individuals aged 50 years or older (*Chemaitelly et al., 2021b*). Age served as the primary eligibility criterion throughout the campaign (*Chemaitelly et al., 2021b*). Booster vaccinations were introduced in the fall of 2021, following a similar prioritization plan (*Figure 1—figure supplement 1B*; *Abu-Raddad et al., 2022a*). However, with the increased availability of vaccine doses, eligibility for boosters was rapidly expanded to include all adults.

### Study design

Two national, matched, retrospective cohort studies were conducted to investigate the potential for indication effect or healthy vaccinee effect influencing estimated effectiveness of COVID-19 primary series (two-dose) and booster (three-dose) vaccinations in Qatar's population. Given the objective of exploring these effects, the studies were designed as vaccine effectiveness studies, adhering to cohort designs developed and implemented in Qatar's population since the pandemic's onset (*Abu-Raddad et al., 2022a*; *Abu-Raddad et al., 2021b*; *Chemaitelly et al., 2022b*; *Chemaitelly et al., 2023a*; *Chemaitelly et al., 2023d*; *Chemaitelly et al., 2023c*; *Mahmoud et al., 2023*).

The healthy vaccinee effect was defined as a bias in vaccine effectiveness studies where healthier individuals are more likely to receive vaccination, even after controlling for differences in health status

based on available data on coexisting conditions. Meanwhile, the indication effect was defined as a bias in vaccine effectiveness studies where individuals with underlying health conditions are more likely to be vaccinated, even after controlling for differences in health status based on available data on coexisting conditions.

In the first study (two-dose analysis), the incidence of non-COVID-19 death in the national cohort of individuals who received the primary series vaccination (designated as the two-dose cohort) was compared with that in the national cohort of unvaccinated individuals (designated as the unvaccinated cohort). In the second study (three-dose analysis), the incidence of non-COVID-19 death in the national cohort of individuals who received a third (booster) dose of vaccination (designated as the three-dose cohort) was compared with that in the two-dose cohort. For both studies, vaccine effectiveness was also estimated by comparing the incidence of SARS-CoV-2 infection and of severe forms of COVID-19 between the study cohorts.

## Severe, critical, and fatal COVID-19

Severe forms of COVID-19 were classified by trained medical personnel independent of the study investigators (*AlNuaimi et al., 2023*; *Chemaitelly et al., 2023e*; *Chemaitelly et al., 2023b*). The classifications were based on individual chart reviews, adhering to the World Health Organization (WHO) guidelines for defining COVID-19 case severity (acute care hospitalization) (*World Health Organization, 2023b*), criticality (intensive care unit hospitalization) (*World Health Organization, 2023b*), and fatality (*World Health Organization, 2023a*) (Section S4) (*AlNuaimi et al., 2023*; *Chemaitelly et al., 2023e*; *Chemaitelly et al., 2023b*).

These evaluations were implemented throughout the pandemic as part of a national protocol, under which every individual with a SARS-CoV-2-positive test and a concurrent COVID-19 hospital admission was assessed for infection severity at regular intervals until discharge or death, regardless of the hospital length of stay (*AlNuaimi et al., 2023*; *Chemaitelly et al., 2023e*; *Chemaitelly et al., 2023b*).

All COVID-19 deaths in Qatar were systematically identified through this protocol, supplemented by a similar protocol applied to all deaths, irrespective of the cause, to determine whether the death met the criteria for classification as a COVID-19 death (*AlNuaimi et al., 2023*; *Chemaitelly et al., 2023e*; *Chemaitelly et al., 2023b*). Deaths in the population that were not classified as COVID-19 were assumed to be non-COVID-19 deaths.

COVID-19 death was defined per WHO classification as "a death resulting from a clinically compatible illness, in a probable or confirmed COVID-19 case, unless there is a clear alternative cause of death that cannot be related to COVID-19 disease (e.g. trauma). There should be no period of complete recovery from COVID-19 between illness and death. A death due to COVID-19 may not be attributed to another disease (e.g. cancer) and should be counted independently of preexisting conditions that are suspected of triggering a severe course of COVID-19" (*World Health Organization, 2023a*).

## Incidence of SARS-CoV-2 infection

Incidence of SARS-CoV-2 infection was defined as any PCR-positive or rapid-antigen-positive test after the start of follow-up, irrespective of symptomatic presentation. Individuals whose infection progressed to severe, critical, or fatal COVID-19 were classified based on their worst outcome, starting with COVID-19 death (*World Health Organization, 2023a*), followed by critical disease (*World Health Organization, 2023b*), and then severe disease (*World Health Organization, 2023b*) (Section S4). Incidence of outcomes of severe forms of COVID-19 was recorded on the date of the SARS-CoV-2-positive test confirming the infection.

## Cohorts' eligibility and matching

Individuals qualified for inclusion in the two-dose cohort if they received two doses of an mRNA vaccine and in the three-dose cohort if they received three doses of an mRNA vaccine. Those who were administered the ChAdOx1 nCoV-19 (AZD1222) vaccine, a small proportion of the population, or the pediatric 10 μg BNT162b2 vaccine were excluded. Individuals qualified for inclusion in the unvaccinated cohort if they had no vaccination record at the start of follow-up.

Cohorts were matched exactly one-to-one by sex, 10-year age group, nationality, exact coexisting conditions (Section S3), and prior documented SARS-CoV-2 infection status (no prior infection, prior

pre-omicron infection, prior omicron infection, or prior pre-omicron and omicron infections). Since prior infection can affect the health status of an individual—potentially leading to conditions such as Long COVID (*Al-Aly et al., 2022*), affecting vaccination uptake (*Nguyen et al., 2021*), or altering protection against subsequent infection and severe COVID-19 (*Abu-Raddad et al., 2021c*; *Chemaitelly et al., 2022c*)—matching by prior infection status was implemented to balance this confounder across cohorts. Prior infections were classified as pre-omicron if they occurred before December 19, 2021, the onset of the omicron wave in Qatar (*Altarawneh et al., 2022b*) and as omicron thereafter.

For the two-dose analysis, individuals who received their second vaccine dose in a specific calendar week in the two-dose cohort were additionally matched to individuals who had a record of a SARS-CoV-2-negative test in that same calendar week in the unvaccinated cohort. This matching approach ensured that matched pairs were present in Qatar during the same time period and were subject to the same vaccination policies and practices at the time of study recruitment. Individuals who were tested after death or who had an unascertained or discrepant death date were excluded.

Similarly, for the three-dose analysis, individuals who received their third vaccine dose in a specific calendar week in the three-dose cohort were matched to individuals who had a record of a SARS-CoV-2-negative test in that same calendar week in the two-dose cohort. Additionally, individuals in the three-dose cohort were matched to individuals in the two-dose cohort by the calendar week of the second vaccine dose. These matching criteria ensured that the paired individuals received their primary series vaccinations at the same time and were present in Qatar during the same period.

Iterative matching was implemented so that, at the start of follow-up, individuals were alive, had maintained their vaccination status, had the same prior infection status as their match, and had no documented SARS-CoV-2 infection within the previous 90 days. The 90-day threshold was used to avoid misclassification of a previous (prolonged) SARS-CoV-2 infection as an incident infection (*Pilz et al., 2022*; *Kojima et al., 2021*; *Chemaitelly et al., 2024*). Consequently, a prior infection was defined as a SARS-CoV-2-positive test that occurred ≥90 days before the start of follow-up.

The above-detailed matching approach aimed to balance observed confounders that could potentially affect the risk of non-COVID-19 death or the risk of infection across the exposure groups (*Abu-Raddad et al., 2021a*; *Coyle et al., 2021*; *Jeremijenko et al., 2021*; *Al Thani et al., 2021*; *AlNuaimi et al., 2023*). The matching factors were selected based on findings from earlier studies on Qatar's population (*Chemaitelly et al., 2021b*; *Abu-Raddad et al., 2022b*).

The matching algorithm was implemented using *ccmatch* command in Stata 18.0 supplemented with conditions to retain only controls that fulfilled the eligibility criteria and was iterated using loops with as many replications as needed until exhaustion (i.e. no more matched pairs could be identified).

According to this study design and matching approach, individuals in the matched unvaccinated cohort in the two-dose analysis may have contributed follow-up time before receiving the primary series vaccination and subsequently contributed follow-up time as part of the two-dose cohort after receiving the primary series vaccination. Similarly, in the three-dose analysis, individuals in the matched two-dose cohort may have contributed follow-up time before receiving the third (booster) dose, as part of the two-dose cohort, and subsequently contributed follow-up time as part of the three-dose cohort after receiving the third dose.

## Cohorts' follow-up

Follow-up started from the calendar date of the second dose in the two-dose analysis and from the calendar date of the third dose in the three-dose analysis. To ensure exchangeability (*Barda et al., 2021*; *Abu-Raddad et al., 2022a*), both members of each matched pair were censored at the earliest occurrence of receiving an additional vaccine dose.

Accordingly, individuals were followed until the first of any of the following events: a documented SARS-CoV-2 infection (irrespective of symptoms), first-dose vaccination for individuals in the unvaccinated cohort (with matched-pair censoring), third-dose vaccination for individuals in the two-dose cohort (with matched-pair censoring), fourth-dose vaccination for individuals in the three-dose cohort (with matched-pair censoring), death, or the administrative end of follow-up at the end of the study.

## Oversight

The institutional review boards at Hamad Medical Corporation and Weill Cornell Medicine-Qatar approved this retrospective study with a waiver of informed consent. The study was reported

according to the Strengthening the Reporting of Observational Studies in Epidemiology (STROBE; Reporting Standards document).

## Statistical analysis

Eligible and matched cohorts were described using frequency distributions and measures of central tendency, and were compared using standardized mean differences (SMDs). An SMD of ≤0.1 indicated adequate matching (*Austin, 2009*). The cumulative incidence of non-COVID-19 death, defined as proportion of individuals at risk whose primary endpoint during follow-up was a non-COVID-19 death, was estimated using the Kaplan-Meier estimator method (*Kaplan and Meier, 1958*).

Incidence rate of non-COVID-19 death in each cohort, defined as number of non-COVID-19 deaths divided by number of person-weeks contributed by all individuals in the cohort, was estimated along with the corresponding 95% confidence interval (CI), using a Poisson log-likelihood regression model with the Stata 18.0 *stptime* command.

Overall adjusted hazard ratio (aHR), comparing incidence of non-COVID-19 death between the cohorts, and corresponding 95% CI, were calculated using Cox regression models with adjustment for the matching factors, via the Stata 18.0 *stcox* command. This adjustment was implemented to ensure precise and unbiased estimation of the standard variance (*Sjölander and Greenland, 2013*). CIs were not adjusted for multiplicity. Schoenfeld residuals and log-log plots for survival curves were used to examine the proportional hazards assumption. An aHR less than 1 indicated evidence of a healthy vaccinee effect. An aHR greater than 1 indicated evidence of an indication effect.

The overall aHR provides a weighted average of the time-varying hazard ratio (*Stensrud and Hernán, 2020*). To explore differences in the risk of non-COVID-19 death over time, the aHR was also estimated by 6-month intervals from the start of follow-up, using separate Cox regressions, with 'failure' restricted to specific time intervals.

Subgroup analyses estimating the overall aHR stratified by age group (<50 years versus ≥50 years), clinical vulnerability status, and prior infection status were also conducted. Individuals were classified as less clinically vulnerable to severe COVID-19 if they were <50 years of age and had one or no coexisting conditions, and as more clinically vulnerable to severe COVID-19 if they were either ≥50 years of age or <50 years of age but with ≥2 coexisting conditions (*Chemaitelly et al., 2023d*; *Chemaitelly et al., 2023e*).

The study analyzed non-COVID-19 mortality in the population of Qatar. However, some deaths may have occurred outside Qatar when expatriates were traveling abroad or had permanently left the country after the start of follow-up. The matching strategy aimed to mitigate any differential effects of these out-of-country deaths on the matched groups, for instance, by matching on a SARS-CoV-2-negative test among controls to ensure their presence in Qatar during the same period.

To assess whether our results could have been affected by bias due to out-of-country deaths or the matching requirement of a SARS-CoV-2-negative test, two sensitivity analyses were conducted: first, by restricting the cohorts to only Qataris, where out-of-country deaths are unlikely, and second, by eliminating the requirement for matching by a SARS-CoV-2-negative test.

Analogous methods were used to compare incidence of SARS-CoV-2 infection and of severe forms of COVID-19 between study cohorts. The overall aHR, comparing incidence of SARS-CoV-2 infection (or severe forms of COVID-19) between study cohorts, was calculated, including an additional adjustment for the testing rate. Vaccine effectiveness against infection and against severe forms of COVID-19, along with the associated 95% CIs, were derived from the aHR as 1-aHR if the aHR was <1, and as 1/aHR-1 if the aHR was ≥1 (*Tseng et al., 2022*; *Chemaitelly et al., 2023d*). This approach ensured a symmetric scale for both negative and positive effectiveness, spanning from −100% to 100%, resulting in a meaningful interpretation of effectiveness, regardless of the value being positive or negative.

Statistical analyses were performed using Stata/SE version 18.0 (Stata Corporation, College Station, TX, USA).

## Results

### Two-dose analysis

*Figure 1—figure supplement 2* illustrates the process of selecting the study cohorts. *Table 1* outlines the cohorts' baseline characteristics. Each matched cohort comprised 812,583 individuals. Median

**Table 1.** Baseline characteristics of the full and matched cohorts for investigating an indication effect or a healthy vaccinee effect among recipients of primary series or booster (third dose) vaccination in Qatar.

| | Two-dose analysis | | | | | | Three-dose analysis | | | | | |
| | Full eligible cohorts | | | Matched cohorts* | | | Full eligible cohorts | | | Matched cohorts† | | |
| Characteristics | Two-dose N=2,168,050 | Unvaccinated N=3,811,694 | SMD‡ | Two-dose N=812,583 | Unvaccinated N=812,583 | SMD‡ | Three-dose N=714,893 | Two-dose N=2,231,443 | SMD‡ | Three-dose N=330,568 | Two-dose N=330,568 | SMD‡ |
|---|---|---|---|---|---|---|---|---|---|---|---|---|
| Median age (IQR)—years | 38 (31–45) | 32 (24–41) | 0.50§ | 34 (28–41) | 33 (27–40) | 0.07§ | 40 (33–49) | 38 (31–45) | 0.21§ | 38 (32–45) | 39 (34–47) | 0.01§ |
| **Age group—no. (%)** | | | | | | | | | | | | |
| 0–19 years | 106,156 (4.9) | 622,215 (16.3) | | 69,673 (8.6) | 69,673 (8.6) | | 33,216 (4.6) | 107,885 (4.8) | | 9,221 (2.8) | 9,221 (2.8) | |
| 20–29 years | 326,484 (15.1) | 909,809 (23.9) | | 191,420 (23.6) | 191,420 (23.6) | | 72,966 (10.2) | 334,458 (15.0) | | 40,015 (12.1) | 40,015 (12.1) | |
| 30–39 years | 809,250 (37.3) | 1,228,030 (32.2) | | 326,985 (40.2) | 326,985 (40.2) | | 239,713 (33.5) | 834,373 (37.4) | | 139,067 (42.1) | 139,067 (42.1) | |
| 40–49 years | 576,564 (26.6) | 660,453 (17.3) | | 158,847 (19.5) | 158,847 (19.5) | | 204,224 (28.6) | 595,300 (26.7) | | 98,080 (29.7) | 98,080 (29.7) | |
| 50–59 years | 244,963 (11.3) | 268,839 (7.1) | | 51,661 (6.4) | 51,661 (6.4) | | 107,990 (15.1) | 252,382 (11.3) | | 36,284 (11.0) | 36,284 (11.0) | |
| 60–69 years | 80,555 (3.7) | 92,395 (2.4) | | 12,014 (1.5) | 12,014 (1.5) | | 43,815 (6.1) | 82,558 (3.7) | | 7,355 (2.2) | 7,355 (2.2) | |
| 70+ years | 24,078 (1.1) | 29,953 (0.8) | 0.58 | 1,983 (0.2) | 1,983 (0.2) | 0.00 | 12,969 (1.8) | 24,487 (1.1) | 0.23 | 546 (0.2) | 546 (0.2) | |
| **Sex** | | | | | | | | | | | | 0.00 |
| Male | 1,599,920 (73.8) | 2,682,394 (70.4) | | 593,856 (73.1) | 593,856 (73.1) | | 467,443 (65.4) | 1,645,973 (73.8) | | 245,116 (74.1) | 245,116 (74.1) | |
| Female | 568,130 (26.2) | 1,129,300 (29.6) | 0.08 | 218,727 (26.9) | 218,727 (26.9) | 0.00 | 247,450 (34.6) | 585,470 (26.2) | 0.18 | 85,452 (25.9) | 85,452 (25.9) | 0.00 |
| **Nationality¶** | | | | | | | | | | | | |
| Bangladeshi | 306,251 (14.1) | 269,021 (7.1) | | 68,102 (8.4) | 68,102 (8.4) | | 66,000 (9.2) | 312,475 (14.0) | | 37,670 (11.4) | 37,670 (11.4) | |
| Egyptian | 106,392 (4.9) | 184,152 (4.8) | | 40,791 (5.0) | 40,791 (5.0) | | 59,691 (8.3) | 109,910 (4.9) | | 18,103 (5.5) | 18,103 (5.5) | |
| Filipino | 201,002 (9.3) | 277,459 (7.3) | | 76,146 (9.4) | 76,146 (9.4) | | 99,405 (13.9) | 209,620 (9.4) | | 40,680 (12.3) | 40,680 (12.3) | |
| Indian | 531,366 (24.5) | 1,074,425 (28.2) | | 268,830 (33.1) | 268,830 (33.1) | | 222,135 (31.1) | 549,694 (24.6) | | 121,774 (36.8) | 121,774 (36.8) | |
| Nepalese | 233,558 (10.8) | 347,108 (9.1) | | 68,279 (8.4) | 68,279 (8.4) | | 28,584 (4.0) | 239,262 (10.7) | | 20,694 (6.3) | 20,694 (6.3) | |
| Pakistani | 103,600 (4.8) | 223,498 (5.9) | | 46,416 (5.7) | 46,416 (5.7) | | 34,161 (4.8) | 106,177 (4.8) | | 14,548 (4.4) | 14,548 (4.4) | |
| Qatari | 195,030 (9.0) | 319,209 (8.4) | | 64,135 (7.9) | 64,135 (7.9) | | 40,519 (5.7) | 199,550 (8.9) | | 23,062 (7.0) | 23,062 (7.0) | |
| Sri Lankan | 75,586 (3.5) | 127,750 (3.4) | | 21,827 (2.7) | 21,827 (2.7) | | 20,759 (2.9) | 77,913 (3.5) | | 10,988 (3.3) | 10,988 (3.3) | |
| Sudanese | 45,213 (2.1) | 78,528 (2.1) | | 17,594 (2.2) | 17,594 (2.2) | | 12,920 (1.8) | 46,586 (2.1) | | 4,140 (1.3) | 4,140 (1.3) | |
| Other nationalities** | 370,052 (17.1) | 910,544 (23.9) | 0.30 | 140,463 (17.3) | 140,463 (17.3) | 0.00 | 130,719 (18.3) | 380,256 (17.0) | 0.39 | 38,909 (11.8) | 38,909 (11.8) | 0.00 |
| **Coexisting conditions** | | | | | | | | | | | | |
| 0 | 1,809,569 (83.5) | 3,352,859 (88.0) | | 746,840 (91.9) | 746,840 (91.9) | | 540,392 (75.6) | 1,860,263 (83.4) | | 311,376 (94.2) | 311,376 (94.2) | |
| 1 | 183,168 (8.4) | 261,898 (6.9) | | 45,414 (5.6) | 45,414 (5.6) | | 78,872 (11.0) | 189,770 (8.5) | | 12,288 (3.7) | 12,288 (3.7) | |
| 2 | 86,673 (4.0) | 102,968 (2.7) | | 13,988 (1.7) | 13,988 (1.7) | | 44,676 (6.2) | 89,926 (4.0) | | 5,049 (1.5) | 5,049 (1.5) | |
| 3 | 39,989 (1.8) | 42,960 (1.1) | | 3,842 (0.5) | 3,842 (0.5) | | 22,684 (3.2) | 41,422 (1.9) | | 1,149 (0.3) | 1,149 (0.3) | |
| 4 | 22,810 (1.1) | 23,715 (0.6) | | 1,602 (0.2) | 1,602 (0.2) | | 13,504 (1.9) | 23,539 (1.1) | | 558 (0.2) | 558 (0.2) | |
| 5 | 13,035 (0.6) | 13,575 (0.4) | | 657 (0.1) | 657 (0.1) | | 7,590 (1.1) | 13,415 (0.6) | | 122 (<0.01) | 122 (<0.01) | |
| ≥6 | 12,806 (0.6) | 13,719 (0.4) | 0.14 | 240 (<0.01) | 240 (<0.01) | 0.00 | 7,175 (1.0) | 13,108 (0.6) | 0.20 | 26 (<0.01) | 26 (<0.01) | 0.00 |
| **Prior infection status††** | | | | | | | | | | | | |

*Table 1 continued on next page*

*Table 1 continued*

| | Two-dose analysis | | | | | | Three-dose analysis | | | | | |
|---|---|---|---|---|---|---|---|---|---|---|---|---|
| | Full eligible cohorts | | | Matched cohorts* | | | Full eligible cohorts | | | Matched cohorts† | | |
| | Two-dose | Unvaccinated | | Two-dose | Unvaccinated | | Three-dose | Two-dose | | Three-dose | Two-dose | |
| Characteristics | N=2,168,050 | N=3,811,694 | SMD‡ | N=812,583 | N=812,583 | SMD‡ | N=714,893 | N=2,231,443 | SMD‡ | N=330,568 | N=330,568 | SMD‡ |
| No prior infection | 1,957,313 (90.3) | – | | 764,366 (94.1) | 764,366 (94.1) | | 591,083 (82.7) | – | | 287,773 (87.1) | 287,773 (87.1) | |
| Prior pre-omicron infection | 208,058 (9.6) | – | | 46,631 (5.7) | 46,631 (5.7) | | 96,567 (13.5) | – | | 33,864 (10.2) | 33,864 (10.2) | |
| Prior omicron infection | 2,463 (0.1) | – | | 1,548 (0.2) | 1,548 (0.2) | | 24,690 (3.5) | – | | 8,624 (2.6) | 8,624 (2.6) | |
| Prior pre-omicron and omicron infections | 216 (<0.01) | – | – | 38 (<0.01) | 38 (<0.01) | 0.00 | 2,553 (0.4) | – | – | 307 (0.1) | 307 (0.1) | 0.00 |

IQR, interquartile range; SMD, standardized mean difference.

*Cohorts were matched exactly one-to-one by sex, 10-year age group, nationality, type of coexisting conditions, and prior infection status. Persons who received their second vaccine dose in a specific calendar week in the two-dose cohort were additionally matched to persons who had a record for a SARS-CoV-2-negative test in that same calendar week in the unvaccinated cohort, to ensure that matched pairs had presence in Qatar over the same time period.

†Cohorts were matched exactly one-to-one by sex, 10-year age group, nationality, type of coexisting conditions, prior infection status, and calendar week of the second vaccine dose. Persons who received their third vaccine dose in a specific calendar week in the three-dose cohort were additionally matched to persons who had a record for a SARS-CoV-2-negative test in that same calendar week in the two-dose cohort, to ensure that matched pairs had presence in Qatar over the same time period.

‡SMD is the difference in the mean of a covariate between groups divided by the pooled standard deviation. An SMD≤0.1 indicates adequate matching.

§SMD is for the mean difference between groups divided by the pooled standard deviation.

¶Nationalities were chosen to represent the most populous groups in Qatar.

**These comprise up to 183 other nationalities in the unmatched and 148 other nationalities in the matched two-dose analyses, and up to 169 other nationalities in the unmatched and 111 other nationalities in the matched three-dose analyses.

††Ascertained at the start of follow-up. Accordingly, distribution is not available for the unmatched unvaccinated cohort in the two-dose analysis and unmatched two-dose cohort in the three-dose analysis, as the start of follow-up for each person in these reference/control cohorts is determined by that of their match after the matching process is completed.

date of the second vaccine dose was June 21, 2021, for the two-dose cohort. Median duration of follow-up was 206 days (interquartile range [IQR], 41–925 days) in the two-dose cohort and 199 days (IQR, 36–933 days) in the unvaccinated cohort (*Figure 1A*).

During follow-up, 237 non-COVID-19 deaths occurred in the two-dose cohort compared to 306 in the unvaccinated cohort (*Table 2A* and *Figure 1—figure supplement 2*). There were 54,427 SARS-CoV-2 infections recorded in the two-dose cohort, of which 23 progressed to severe, 6 to critical, and none to fatal COVID-19. Meanwhile, 57,974 SARS-CoV-2 infections were recorded in the unvaccinated cohort, of which 539 progressed to severe, 66 to critical, and 25 to fatal COVID-19.

The cumulative incidence of non-COVID-19 death was 0.070% (95% CI: 0.061–0.081%) for the two-dose cohort and 0.071% (95% CI: 0.062–0.080%) for the unvaccinated cohort after 990 days of follow-up (*Figure 1A*). The overall aHR comparing the incidence of non-COVID-19 death in the two-dose cohort to that in the unvaccinated cohort was 0.76 (95% CI: 0.64–0.90), indicating evidence of a healthy vaccinee effect (*Table 2A*).

In the first 6 months of follow-up, the aHR was 0.35 (95% CI: 0.27–0.46), indicating strong evidence of a healthy vaccinee effect (*Figure 2A*). However, the combined analysis of all periods after the first 6 months showed an aHR of 1.52 (95% CI: 1.19–1.94).

The subgroup analyses estimated the aHR at 0.89 (95% CI: 0.72–1.11) among individuals under 50 years of age and at 0.56 (95% CI: 0.42–0.75) among those 50 years of age and older (*Table 3A*). The aHR was 0.98 (95% CI: 0.79–1.22) for those less clinically vulnerable to severe COVID-19 and 0.51 (95% CI: 0.39–0.68) for the more clinically vulnerable group. The aHR by prior infection status was 0.74 (95% CI: 0.63–0.89) for no prior infection and 1.00 (95% CI: 0.45–2.20) for prior pre-omicron infection.

In the two sensitivity analyses—one including only Qataris and the other also including only Qataris but without matching on a SARS-CoV-2-negative test among controls—the aHRs for non-COVID-19 death were 0.29 (95% CI: 0.19–0.43) and 0.38 (95% CI: 0.30–0.50), respectively (*Table 4A*). Both analyses are consistent with each other and with the main analysis results (*Table 4*). However, the healthy vaccinee effect is more pronounced among Qataris, as the proportion of individuals above 50 years of age or those with serious coexisting conditions is substantially higher among Qataris compared to the

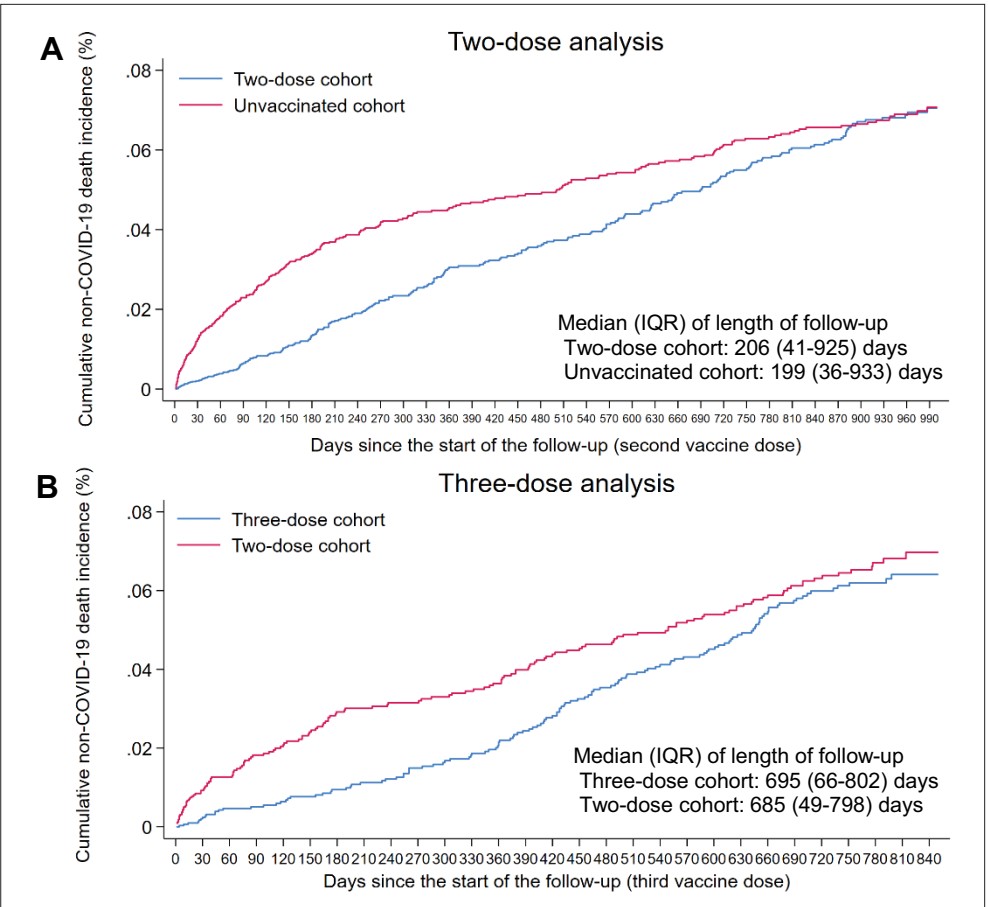

**Figure 1.** Cumulative incidence of non-COVID-19 death in the matched (**A**) two-dose cohort compared to the unvaccinated cohort and (**B**) three-dose cohort compared to the two-dose cohort.

The online version of this article includes the following source data and figure supplement(s) for figure 1:

**Source data 1.** Data used to generate *Figure 1A and B*.

**Figure supplement 1.** Distribution of vaccinations.

**Figure supplement 2.** Flowchart describing the study population selection process for investigating an indication effect or a healthy vaccinee effect among recipients of primary series vaccination compared to those with no vaccination in Qatar.

**Figure supplement 3.** Flowchart describing the study population selection process for investigating an indication effect or a healthy vaccinee effect among recipients of booster (third dose) vaccination compared to recipients of primary series vaccination in Qatar.

rest of the population, which primarily comprises working-age male craft and manual workers (*Abu-Raddad et al., 2021a*; *Al Thani et al., 2021*; *AlNuaimi et al., 2023*).

The overall effectiveness of primary series vaccination compared to no vaccination was 10.7% (95% CI: 9.6–11.7) against infection and 95.9% (95% CI: 94.0–97.1) against severe, critical, or fatal COVID-19 (*Table 2A*). *Figure 2—figure supplement 1* further illustrates the effectiveness of primary series vaccination against severe, critical, or fatal COVID-19, stratified by time since vaccination, both overall and within the subgroups of individuals aged <50 years and those aged ≥50 years.

## Three-dose analysis

*Figure 1—figure supplement 3* illustrates the process of selecting the study cohorts. *Table 1* outlines the cohorts' baseline characteristics. Each matched cohort comprised 330,568 individuals. The median date of the second vaccine dose was May 15, 2021, for both the two-dose and three-dose cohorts. The median date of the third vaccine dose in the three-dose cohort was January 24, 2022. The median

**Table 2.** Hazard ratios for incidence of non-COVID-19 death, SARS-CoV-2 infection, and severe, critical, or fatal COVID-19 in the (A) two-dose analysis and (B) three-dose analysis.

| (A) Two-dose analysis | Two-dose cohort* | Unvaccinated cohort* |
|---|---|---|
| Sample size | 812,583 | 812,583 |
| Number of non-COVID-19 death | 237 | 306 |
| Number of incident infections | 54,427 | 57,974 |
| Number of severe, critical, or fatal COVID-19 disease | 29 | 630 |
| Total follow-up time (person-weeks) | 46,028,318 | 46,275,391 |
| **Non-COVID-19 death** | | |
| Incidence rate of non-COVID-19 death (per 10,000 person-weeks; 95% CI) | 0.05 (0.05–0.06) | 0.07 (0.06–0.07) |
| Unadjusted hazard ratio for non-COVID-19 death (95% CI) | 0.77 (0.65–0.91) | |
| Adjusted hazard ratio for non-COVID-19 death (95% CI)[†] | 0.76 (0.64–0.90) | |
| **SARS-CoV-2 infection** | | |
| Unadjusted hazard ratio for SARS-CoV-2 infection (95% CI) | 0.93 (0.92–0.94) | |
| Adjusted hazard ratio for SARS-CoV-2 infection (95% CI)[‡] | 0.89 (0.88–0.90) | |
| Effectiveness against SARS-CoV-2 infection (95% CI)[‡] | 10.7 (9.6–11.7) | |
| **Severe, critical, or fatal COVID-19 disease** | | |
| Unadjusted hazard ratio for severe, critical, or fatal COVID-19 disease (95% CI) | 0.05 (0.03–0.07) | |
| Adjusted hazard ratio for severe, critical, or fatal COVID-19 disease (95% CI)[‡] | 0.04 (0.03–0.06) | |
| Effectiveness against severe, critical, or fatal COVID-19 disease (95% CI)[‡] | 95.9 (94.0–97.1) | |
| (B) Three-dose analysis | Three-dose cohort[§] | Two-dose cohort[§] |
| Sample size | 330,568 | 330,568 |
| Number of non-COVID-19 death | 132 | 147 |
| Number of incident infections | 26,842 | 35,411 |
| Number of severe, critical, or fatal COVID-19 disease | 6 | 9 |
| Total follow-up time (person-weeks) | 24,015,307 | 23,088,912 |
| **Non-COVID-19 death** | | |
| Incidence rate of non-COVID-19 death (per 10,000 person-weeks; 95% CI) | 0.05 (0.05–0.07) | 0.06 (0.05–0.07) |
| Unadjusted hazard ratio for non-COVID-19 death (95% CI)[¶] | 0.87 (0.68–1.10) | |
| Adjusted hazard ratio for non-COVID-19 death (95% CI)[¶] | 0.85 (0.67–1.07) | |
| **SARS-CoV-2 infection** | | |
| Unadjusted hazard ratio for SARS-CoV-2 infection (95% CI) | 0.74 (0.72–0.75) | |
| Adjusted hazard ratio for SARS-CoV-2 infection (95% CI)** | 0.74 (0.72–0.75) | |
| Effectiveness against SARS-CoV-2 infection (95% CI)** | 26.3 (25.2–27.5) | |
| **Severe, critical, or fatal COVID-19 disease** | | |
| Unadjusted hazard ratio for severe, critical, or fatal COVID-19 disease (95% CI) | 0.64 (0.23–1.81) | |
| Adjusted hazard ratio for severe, critical, or fatal COVID-19 disease (95% CI)** | 0.66 (0.23–1.86) | |
| Effectiveness against severe, critical, or fatal COVID-19 disease (95% CI)** | 34.1 (−46.4–76.7) | |

*Table 2 continued on next page*

*Table 2 continued*

CI, confidence interval; COVID-19, coronavirus disease 2019; SARS-CoV-2, severe acute respiratory syndrome coronavirus 2.

*Cohorts were matched exactly one-to-one by sex, 10-year age group, nationality, type of coexisting conditions, and prior infection status. Persons who received their second vaccine dose in a specific calendar week in the two-dose cohort were additionally matched to persons who had a record for a SARS-CoV-2-negative test in that same calendar week in the unvaccinated cohort, to ensure that matched pairs had presence in Qatar over the same time period.

†Adjusted for sex, 10-year age group, nationality, number of coexisting conditions, prior infection status, and calendar week of the second vaccine dose for the two-dose cohort or SARS-CoV-2-negative test for the unvaccinated cohort.

‡Adjusted for sex, 10-year age group, nationality, number of coexisting conditions, prior infection status, calendar week of the second vaccine dose for the two-dose cohort or SARS-CoV-2-negative test for the unvaccinated cohort, and testing rate.

§Cohorts were matched exactly one-to-one by sex, 10-year age group, nationality, type of coexisting conditions, prior infection status, and calendar week of the second vaccine dose. Persons who received their third vaccine dose in a specific calendar week in the three-dose cohort were additionally matched to persons who had a record for a SARS-CoV-2-negative test in that same calendar week in the two-dose cohort, to ensure that matched pairs had presence in Qatar over the same time period.

¶Adjusted for sex, 10-year age group, nationality, number of coexisting conditions, prior infection status, and calendar week of the second vaccine dose.

**Adjusted for sex, 10-year age group, nationality, number of coexisting conditions, prior infection status, calendar week of the second vaccine dose, and testing rate.

duration of follow-up was 695 days (IQR, 66–802 days) in the three-dose cohort and 685 days (IQR, 49–798 days) in the two-dose cohort (*Figure 1B*).

During follow-up, 132 non-COVID-19 deaths occurred in the three-dose cohort compared to 147 in the two-dose cohort (*Table 2B* and *Figure 1—figure supplement 3*). There were 26,842 SARS-CoV-2 infections recorded in the three-dose cohort, of which 3 progressed to severe, 2 to critical, and 1 to fatal COVID-19. Meanwhile, 35,411 SARS-CoV-2 infections were recorded in the two-dose cohort, of which 8 progressed to severe, 1 to critical, and none to fatal COVID-19.

The cumulative incidence of non-COVID-19 death was 0.064% (95% CI: 0.054–0.076%) for the three-dose cohort and 0.070% (95% CI: 0.059–0.083%) for the two-dose cohort, after 840 days of follow-up (*Figure 1B*). The overall aHR comparing the incidence of non-COVID-19 death in the three-dose cohort to that in the two-dose cohort was 0.85 (95% CI: 0.67–1.07), indicating no overall evidence of a healthy vaccinee effect (*Table 2B*).

In the first 6 months of follow-up, the aHR was 0.31 (95% CI: 0.20–0.50), indicating strong evidence of a healthy vaccinee effect (*Figure 2B*). However, the combined analysis of all subsequent periods showed an aHR of 1.37 (95% CI: 1.02–1.85).

The subgroup analyses estimated the aHR at 0.90 (95% CI: 0.67–1.20) among individuals under 50 years of age and at 0.76 (95% CI: 0.51–1.13) among those 50 years of age and older (*Table 3B*). The aHR was 0.91 (95% CI: 0.67–1.22) for those less clinically vulnerable to severe COVID-19 and 0.76 (95% CI: 0.52–1.12) for the more clinically vulnerable group. The aHR by prior infection status was 0.79 (95% CI: 0.61–1.01) for no prior infection, 1.63 (95% CI: 0.71–3.72) for prior pre-omicron infection, and 1.32 (95% CI: 0.30–5.91) for prior omicron infection.

In the two sensitivity analyses—one including only Qataris and the other also including only Qataris but without matching on a SARS-CoV-2-negative test among controls—the aHRs for non-COVID-19 death were 0.76 (95% CI: 0.43–1.32) and 0.77 (95% CI: 0.53–1.13), respectively (*Table 4B*). Both analyses are consistent with each other and with the main analysis results.

The overall effectiveness of the third-dose (booster) vaccination compared to the primary series vaccination against infection was 26.3% (95% CI: 25.2–27.5) (*Table 2B*). However, no significant effect of the third dose was observed against severe, critical, or fatal COVID-19 (34.1%; 95% CI: –46.4 to 76.7). *Figure 2—figure supplement 1* further illustrates the overall effectiveness of the third-dose vaccination against severe, critical, or fatal COVID-19, stratified by time since vaccination.

## Discussion

The results confirm the presence of a healthy vaccinee effect in rigorously conducted vaccine effectiveness studies. Despite meticulous cohort matching, a particularly pronounced healthy vaccinee effect was evident during the first 6 months after vaccination. Notably, the same effect, with a similar magnitude, was observed in both primary series and booster vaccinations, suggesting a consistent underlying phenomenon.

This effect, similar to that found in influenza vaccine effectiveness studies (*Jackson et al., 2006a*; *Nelson et al., 2009*), may stem from seriously ill and end-of-life individuals, such as terminal cancer patients, as well as frail and less mobile elderly persons, being less likely to be vaccinated (*Jackson*

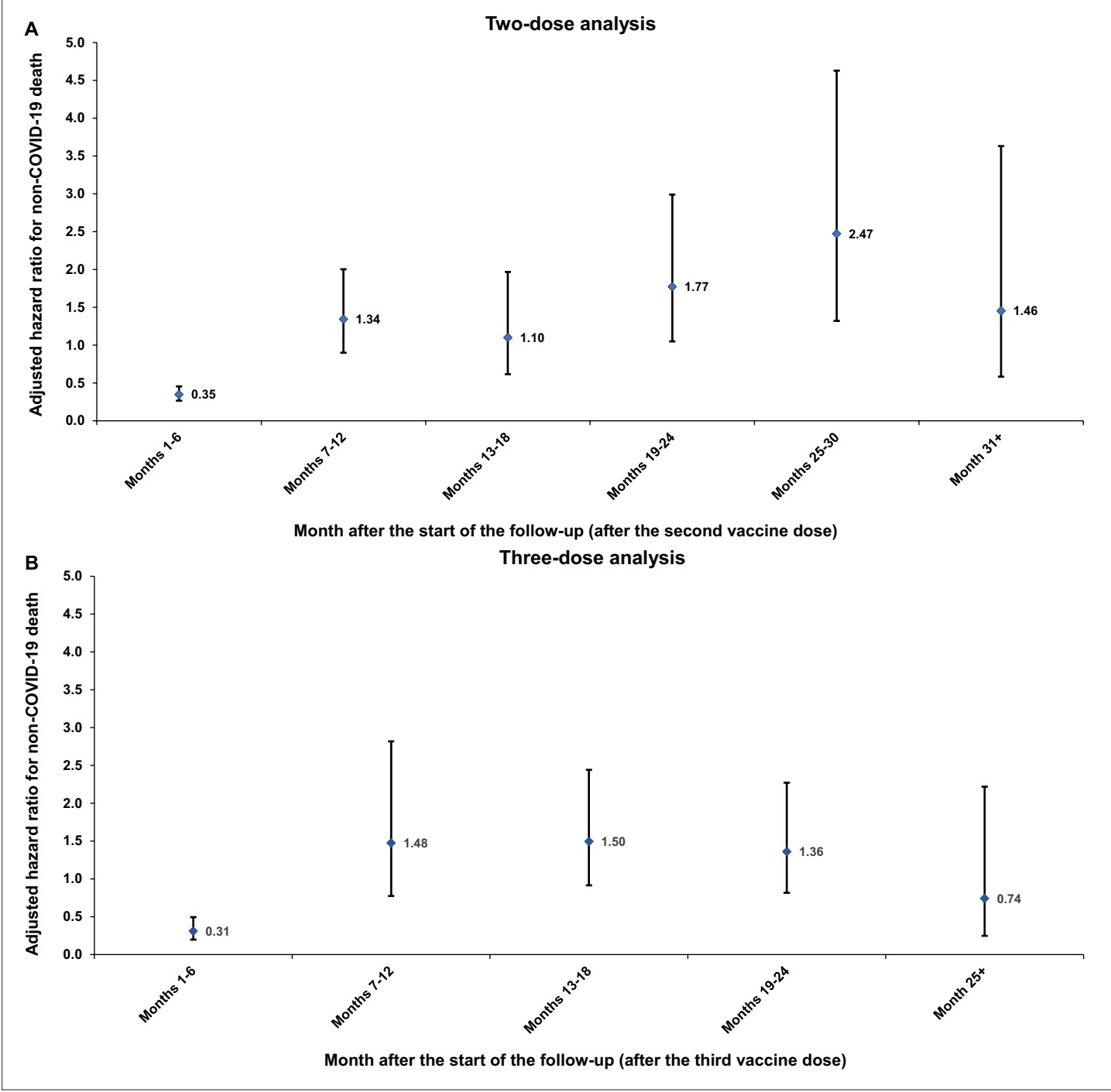

**Figure 2.** Adjusted hazard ratios for incidence of non-COVID-19 death in the (**A**) two-dose analysis and (**B**) three-dose analysis, by 6-month interval of follow-up. Error bars indicate the corresponding 95% confidence intervals.

The online version of this article includes the following source data and figure supplement(s) for figure 2:

**Source data 1.** Data used to generate *Figure 2A and B*.

**Figure supplement 1.** Adjusted hazard ratios for incidence of severe, critical, or fatal COVID-19 in the two-dose and three-dose analyses, by 6-month interval of follow-up.

**Table 3.** Subgroup analyses.

Hazard ratios for incidence of non-COVID-19 death stratified by age group, clinical vulnerability status, and prior infection status in the (A) two-dose analysis and (B) three-dose analysis.

| (A) Two-dose analysis | Two-dose cohort* | Unvaccinated cohort* |
|---|---|---|
| **Age** | | |
| **<50 years of age** | | |
| Unadjusted hazard ratio for non-COVID-19 death (95% CI) | 0.91 (0.73–1.12) | |
| Adjusted hazard ratio for non-COVID-19 death (95% CI)[†] | 0.89 (0.72–1.11) | |
| **≥50 years of age** | | |
| Unadjusted hazard ratio for non-COVID-19 death (95% CI) | 0.59 (0.44–0.78) | |
| Adjusted hazard ratio for non-COVID-19 death (95% CI)[†] | 0.56 (0.42–0.75) | |
| **Clinical vulnerability status** | | |
| **Less clinically vulnerable to severe COVID-19** | | |
| Unadjusted hazard ratio for non-COVID-19 death (95% CI) | 0.99 (0.80–1.23) | |
| Adjusted hazard ratio for non-COVID-19 death (95% CI)[†] | 0.98 (0.79–1.22) | |
| **More clinically vulnerable to severe COVID-19** | | |
| Unadjusted hazard ratio for non-COVID-19 death (95% CI) | 0.53 (0.41–0.70) | |
| Adjusted hazard ratio for non-COVID-19 death (95% CI)[†] | 0.51 (0.39–0.68) | |
| **Prior infection status** | | |
| **No prior infection** | | |
| Unadjusted hazard ratio for non-COVID-19 death (95% CI) | 0.76 (0.64–0.90) | |
| Adjusted hazard ratio for non-COVID-19 death (95% CI)[†] | 0.74 (0.63–0.89) | |
| **Prior pre-omicron infection** | | |
| Unadjusted hazard ratio for non-COVID-19 death (95% CI) | 1.05 (0.48–2.30) | |
| Adjusted hazard ratio for non-COVID-19 death (95% CI)[†] | 1.00 (0.45–2.20) | |
| **Prior omicron infection** | | |
| Unadjusted hazard ratio for non-COVID-19 death (95% CI) | -- [‡] | |
| Adjusted hazard ratio for non-COVID-19 death (95% CI)[†] | -- [‡] | |
| **Prior pre-omicron & omicron infections** | | |
| Unadjusted hazard ratio for non-COVID-19 death (95% CI) | -- [‡] | |
| Adjusted hazard ratio for non-COVID-19 death (95% CI)[†] | -- [‡] | |
| **(B) Three-dose analysis** | **Three-dose cohort [§]** | **Two-dose cohort [§]** |
| **Age** | | |
| **<50 years of age** | | |
| Unadjusted hazard ratio for non-COVID-19 death (95% CI) | 0.90 (0.67–1.21) | |
| Adjusted hazard ratio for non-COVID-19 death (95% CI)[¶] | 0.90 (0.67–1.20) | |
| **≥50 years of age** | | |
| Unadjusted hazard ratio for non-COVID-19 death (95% CI) | 0.80 (0.54–1.18) | |
| Adjusted hazard ratio for non-COVID-19 death (95% CI)[¶] | 0.76 (0.51–1.13) | |
| **Clinical vulnerability status** | | |
| **Less clinically vulnerable to severe COVID-19** | | |

*Table 3 continued on next page*

*Table 3 continued*

| (B) Three-dose analysis | Three-dose cohort [§] Two-dose cohort [§] |
|---|---|
| Unadjusted hazard ratio for non-COVID-19 death (95% CI) | 0.91 (0.68–1.23) |
| Adjusted hazard ratio for non-COVID-19 death (95% CI)[¶] | 0.91 (0.67–1.22) |
| **More clinically vulnerable to severe COVID-19** | |
| Unadjusted hazard ratio for non-COVID-19 death (95% CI) | 0.78 (0.54–1.15) |
| Adjusted hazard ratio for non-COVID-19 death (95% CI)[¶] | 0.76 (0.52–1.12) |
| Prior infection status | |
| No prior infection | |
| Unadjusted hazard ratio for non-COVID-19 death (95% CI) | 0.80 (0.63–1.03) |
| Adjusted hazard ratio for non-COVID-19 death (95% CI)[¶] | 0.79 (0.61–1.01) |
| Prior pre-omicron infection | |
| Unadjusted hazard ratio for non-COVID-19 death (95% CI) | 1.63 (0.71–3.73) |
| Adjusted hazard ratio for non-COVID-19 death (95% CI)[¶] | 1.63 (0.71–3.72) |
| Prior omicron infection | |
| Unadjusted hazard ratio for non-COVID-19 death (95% CI) | 1.32 (0.30–5.90) |
| Adjusted hazard ratio for non-COVID-19 death (95% CI)[¶] | 1.32 (0.30–5.91) |
| Prior pre-omicron & omicron infections | |
| Unadjusted hazard ratio for non-COVID-19 death (95% CI) | -- [‡] |
| Adjusted hazard ratio for non-COVID-19 death (95% CI)[¶] | -- [‡] |

CI, confidence interval; COVID-19, coronavirus disease 2019; SARS-CoV-2, severe acute respiratory syndrome coronavirus 2.

*Cohorts were matched exactly one-to-one by sex, 10-year age group, nationality, type of coexisting conditions, and prior infection status. Persons who received their second vaccine dose in a specific calendar week in the two-dose cohort were additionally matched to persons who had a record for a SARS-CoV-2-negative test in that same calendar week in the unvaccinated cohort, to ensure that matched pairs had presence in Qatar over the same time period.

[†]Adjusted for sex, 10-year age group, nationality, number of coexisting conditions, prior infection status (where applicable), and calendar week of the second vaccine dose for the two-dose cohort or SARS-CoV-2-negative test for the unvaccinated cohort.

[‡]Could not be estimated because of no or small number of events.

[§]Cohorts were matched exactly one-to-one by sex, 10-year age group, nationality, type of coexisting conditions, prior infection status, and calendar week of the second vaccine dose. Persons who received their third vaccine dose in a specific calendar week in the three-dose cohort were additionally matched to persons who had a record for a SARS-CoV-2-negative test in that same calendar week in the two-dose cohort, to ensure that matched pairs had presence in Qatar over the same time period.

[¶]Adjusted for sex, 10-year age group, nationality, number of coexisting conditions, prior infection status (where applicable), and calendar week of the second vaccine dose.

*et al., 2006a*; *Nelson et al., 2009*). This leads to a higher short-term mortality risk among the unvaccinated (*Jackson et al., 2006a*; *Nelson et al., 2009*). This is supported by this effect being only evident in the first 6 months, and specifically among those aged 50 years and older and those more clinically vulnerable to severe COVID-19. Given the strength of this effect, it seems unlikely that it can be attributed to an effect of vaccination-induced nonspecific immune activation or trained/bystander immunity that protects against a range of infectious and noninfectious outcomes (*Benn et al., 2013*; *Netea et al., 2020*; *Xu et al., 2023*; *Tayar et al., 2023*).

These findings raise a concern, as vaccine effectiveness is typically estimated for the first few months after vaccination for seasonal infections, or for infections with repeated waves, such as influenza and SARS-CoV-2 (*Jackson et al., 2006a*; *Nelson et al., 2009*; *Remschmidt et al., 2015*; *Feikin et al., 2022*). The findings support a rationale for excluding seriously ill, immunosuppressed, or functionally

**Table 4.** Sensitivity analyses.

Hazard ratios for incidence of non-COVID-19 death among Qataris with and without matching on a SARS-CoV-2-negative test among controls in the (A) two-dose analysis and (B) three-dose analysis.

| (A) Two-dose analysis | Two-dose cohort | Unvaccinated cohort |
|---|---|---|
| Sensitivity analysis I-Restricting analysis to Qataris[*] | | |
| Unadjusted hazard ratiofor non-COVID-19 death (95% CI) | 0.29 (0.19–0.43) | |
| Adjusted hazard ratiofor non-COVID-19 death (95% CI)[†] | 0.29 (0.19–0.43) | |
| Sensitivity analysis II-Restricting analysis to Qataris and not matching by a SARS-CoV-2-negative test among controls[‡] | | |
| Unadjusted hazard ratiofor non-COVID-19 death (95% CI) | 0.40 (0.31–0.51) | |
| Adjusted hazard ratiofor non-COVID-19 death (95% CI)[§] | 0.38 (0.30–0.50) | |
| **(B) Three-dose analysis** | **Three-dose cohort** | **Two-dose cohort** |
| Sensitivity analysis I-Restricting analysis to Qataris[¶] | | |
| Unadjusted hazard ratiofor non-COVID-19 death (95% CI) | 0.77 (0.44–1.33) | |
| Adjusted hazard ratiofor non-COVID-19 death (95% CI)[**] | 0.76 (0.43–1.32) | |
| Sensitivity analysis II-Restricting analysis to Qataris and not matching by a SARS-CoV-2-negative test among controls[††] | | |
| Unadjusted hazard ratiofor non-COVID-19 death (95% CI) | 0.77 (0.52–1.12) | |
| Adjusted hazard ratiofor non-COVID-19 death (95% CI)[**] | 0.77 (0.53–1.13) | |

CI, confidence interval; COVID-19, coronavirus disease 2019; SARS-CoV-2, severe acute respiratory syndrome coronavirus 2.

[*]Cohorts were matched exactly one-to-one by sex, 10-year age group, type of coexisting conditions, and prior infection status. Persons who received their second vaccine dose in a specific calendar week in the two-dose cohort were additionally matched to persons who had a record for a SARS-CoV-2-negative test in that same calendar week in the unvaccinated cohort, to ensure that matched pairs had presence in Qatar over the same time period.

[†]Adjusted for sex, 10-year age group, number of coexisting conditions, prior infection status, and calendar week of the second vaccine dose for the two-dose cohort or SARS-CoV-2-negative test for the unvaccinated cohort.

[‡]Cohorts were matched exactly one-to-one by sex, 10-year age group, type of coexisting conditions, and prior infection status.

[§]Adjusted for sex, 10-year age group, number of coexisting conditions, and prior infection status.

[¶]Cohorts were matched exactly one-to-one by sex, 10-year age group, type of coexisting conditions, prior infection status, and calendar week of the second vaccine dose. Persons who received their third vaccine dose in a specific calendar week in the three-dose cohort were additionally matched to persons who had a record for a SARS-CoV-2-negative test in that same calendar week in the two-dose cohort, to ensure that matched pairs had presence in Qatar over the same time period.

[**]Adjusted for sex, 10-year age group, number of coexisting conditions, prior infection status, and calendar week of the second vaccine dose.

[††]Cohorts were matched exactly one-to-one by sex, 10-year age group, type of coexisting conditions, prior infection status, and calendar week of the second vaccine dose.

impaired individuals in studies of vaccine effectiveness in the general population (*Jackson et al., 2006a*; *Nelson et al., 2009*).

While the mortality risk was higher among the unvaccinated group during the first 6 months after vaccination, it subsequently reversed, becoming higher among the vaccinated group. This reversal may be attributed to the depletion of seriously ill individuals in the unvaccinated group during the initial 6 months, leaving behind a cohort enriched with relatively healthier individuals.

While the healthy vaccinee effect was evident, it is plausible that both a healthy vaccinee effect and an indication effect could coexist, manifesting at different strengths and time points. The healthy vaccinee effect is likely driven by seriously ill or end-of-life individuals who are less likely to seek vaccination. In contrast, the indication effect could be driven by individuals with serious but less immediately life-threatening conditions, such as heart disease or diabetes, who pursue vaccination to mitigate their heightened risk of severe infection or complications related to these conditions. Such health

conditions elevate mortality risk over a longer time horizon rather than immediately. Consequently, the higher mortality risk observed beyond 6 months is not inconsistent with a potential presence of an indication effect.

Although a healthy vaccinee effect was observed in this study, the extent to which this effect may have biased and skewed the estimated vaccine effectiveness remains uncertain. This effect is presumably more likely to bias vaccine effectiveness against severe forms of COVID-19 than against infection alone (*Høeg et al., 2023*). The impact of this effect might also have been mitigated somewhat by using specific infection outcomes—such as severe, critical, or fatal COVID-19—and by confirming infections through laboratory methods, rather than relying on broad nonspecific outcomes like all-cause mortality, commonly used in influenza vaccine effectiveness studies (*Jackson et al., 2006a*; *Nelson et al., 2009*; *Remschmidt et al., 2015*). Ironically, the overall healthy vaccinee effect over the entire duration of follow-up may have been partially mitigated by an indication effect.

The results indicated strong protection from vaccination against severe forms of COVID-19, with an observed effectiveness of 96% for the primary series. However, vaccine effectiveness against infection was modest, which is expected given that this type of protection rapidly diminishes within the first few months after vaccination (*Feikin et al., 2022*; *Abu-Raddad et al., 2022c*; *Chemaitelly et al., 2021b*), and effectiveness was estimated over 3 years of follow-up.

This study has limitations, which were assessed within the context of potential risk of bias in nonrandomized studies of interventions (*Sterne et al., 2016*), drawing on prior literature related to the investigated effects and our previous work using similar study designs on these national databases.

While all COVID-19-related deaths in Qatar were systematically identified through national protocols, as described in the Methods, and made available to the study investigators, the specific causes of non-COVID-19 deaths were not accessible. This limitation constrained the scope of additional analyses that could have been conducted. As a result, while this study provides evidence of the healthy vaccinee effect in rigorously conducted vaccine effectiveness studies, characterizing both its effect size and temporal profile, it does not identify the specific cause of this bias.

Further research is needed to investigate this bias by collecting primary data on the relationship between comorbidity and frailty and vaccination behavior. For example, while older, independent, and active community members may demonstrate a greater preference for vaccination, this may not hold true for frail residents in long-term care facilities. However, such long-term care facilities may enforce a policy of vaccinating all residents. It is also important to examine whether this relationship varies across different vaccine types, such as COVID-19 and influenza.

A number of non-COVID-19 deaths had unascertained or discrepant death date; therefore, these individuals were excluded from the study from the onset. However, this exclusion is not likely to materially affect the analyses, as there were only 23 deaths with unascertained or discrepant death dates in the entire population of Qatar over the 3 years of this study.

Documented COVID-19 deaths may not include all deaths that occurred because of COVID-19 (*Islam et al., 2021*; *Kontis et al., 2020*), and thus there could be some misclassification bias affecting the distinction between COVID-19 and non-COVID-19 deaths. However, the number of COVID-19 deaths was small (*Figure 1—figure supplement 2* and *Figure 1—figure supplement 3*), and the COVID-19 death rate in the young and working-age population of Qatar has been one of the lowest worldwide, with less than 0.1% of documented infections resulting in death (*AlNuaimi et al., 2023*; *Chemaitelly et al., 2023e*; *Chemaitelly et al., 2023b*; *Johns Hopkins Coronavirus Resource, 2022*). Earlier studies suggest that the number of undocumented COVID-19 deaths in Qatar is too small to appreciably affect the analyses of this study (*AlNuaimi et al., 2023*; *Chemaitelly et al., 2023e*; *Chemaitelly et al., 2023b*).

The study analyzed all deaths occurring within Qatar; however, some deaths might have occurred outside the country. Data on deaths outside the country were not available for our analysis. Nevertheless, the matching process was designed to ensure that participants were present in Qatar during the same period and to balance the risk of out-of-country deaths across cohorts. Consequently, these out-of-country deaths are not likely to have influenced the comparative outcomes of the matched cohorts. Further supporting our results, the sensitivity analysis, which was restricted to only Qataris—a group very unlikely to experience out-of-Qatar deaths—corroborated the main study results.

The national testing database served as a sampling frame for unvaccinated individuals in Qatar. However, this database does not capture individuals who have never had a SARS-CoV-2 test since

the onset of the pandemic. Nevertheless, testing has been extensive in Qatar, with the vast majority conducted for routine reasons (*Altarawneh et al., 2022b*; *Chemaitelly et al., 2021b*). Given the widespread testing mandates and the large volume of tests conducted, it is not likely that any citizen or resident in Qatar has not had at least one SARS-CoV-2 test since the onset of the pandemic (*Altarawneh et al., 2022b*; *Chemaitelly et al., 2021b*).

Matched unvaccinated individuals were required to have tested negative for SARS-CoV-2 in the week their matched vaccinated counterparts received their vaccine, ensuring that both groups were present in Qatar during the same time period. Different eligibility criteria between the two arms could bias the study if there was a correlation between testing and non-COVID-19 death. However, the sensitivity analysis for Qataris, which eliminated the requirement for matching by a SARS-CoV-2-negative test, confirmed similar results, suggesting that this matching requirement may not have biased the results.

A consequence of the rigorous matching employed in this study is that the matched cohorts are not fully representative of the population of Qatar. However, the study was specifically designed to address a focused research question: the existence of the healthy vaccinee effect in rigorously conducted vaccine effectiveness studies. Therefore, the emphasis is on the internal validity of the study—ensuring that the relationship between the exposure and outcome is measured accurately and is free from bias and confounding—rather than on external validity or generating estimates specific to the population of Qatar.

This approach parallels the design of vaccine efficacy RCTs, which are typically conducted in select populations that may not fully represent the broader population (e.g. excluding pregnant women, children, or individuals with certain coexisting conditions). The objective of such RCTs and similarly this study—through the use of rigorous matching to control for selection bias, akin to the role of randomization in an RCT—is not to generate population-wide estimates but rather to provide a precise and unbiased measure of the exposure-outcome relationship under investigation.

The study was conducted in a specific national population consisting mainly of healthy working-age adults, thus the generalizability of the findings to other populations remains uncertain. As an observational study, the investigated cohorts were neither blinded nor randomized, so unmeasured or uncontrolled confounding factors cannot be excluded. Although matching accounted for key factors affecting risks of death and infection (*Abu-Raddad et al., 2021a*; *Coyle et al., 2021*; *Al Thani et al., 2021*; *Jeremijenko et al., 2021*), it was not possible for other factors such as geography or occupation, for which data were unavailable. However, Qatar is essentially a city-state where infection incidence was broadly distributed across neighborhoods. Nearly 90% of Qatar's population are expatriates from over 150 countries, primarily coming for employment (*Abu-Raddad et al., 2021a*). In this context, nationality, age, and sex serve as powerful proxies for socioeconomic status (*Abu-Raddad et al., 2021a*; *Coyle et al., 2021*; *Al Thani et al., 2021*; *Jeremijenko et al., 2021*). Nationality is also strongly associated with occupation (*Abu-Raddad et al., 2021a*; *Coyle et al., 2021*; *Al Thani et al., 2021*; *Jeremijenko et al., 2021*).

The matching procedure used in this study has been evaluated in previous studies with different epidemiologic designs and using control groups to test for null effects (*Abu-Raddad et al., 2021d*; *Chemaitelly et al., 2021b*; *Chemaitelly et al., 2021c*; *Abu-Raddad et al., 2022c*; *Abu-Raddad et al., 2022b*). These prior studies demonstrated that this procedure balances differences in infection exposure to estimate vaccine effectiveness (*Abu-Raddad et al., 2021d*; *Chemaitelly et al., 2021b*; *Chemaitelly et al., 2021c*; *Abu-Raddad et al., 2022c*; *Abu-Raddad et al., 2022b*), suggesting that the matching strategy may also have mitigated differences in mortality risk. Lastly, the aHRs were estimated both overall and by 6-month intervals from the start of follow-up. However, the interval-based analysis can be susceptible to changes in the composition of the study population over time.

The study has strengths. It was implemented on Qatar's entire population and sizable cohorts, representing a diverse range of national backgrounds. Extensive, validated databases from numerous prior COVID-19 studies were utilized in this study. The availability of an integrated digital health information platform provided data on various confounding factors, facilitating rigorous matching based on specific coexisting conditions and prior infection statuses. The ascertainment of COVID-19 deaths was meticulously conducted by trained personnel, adhering to a national protocol and WHO guidelines for classifying COVID-19 case fatalities (*World Health Organization, 2023a*).

In conclusion, a healthy vaccinee effect was observed, but only in the first 6 months following COVID-19 vaccination and specifically among those aged 50 years and older and those more clinically vulnerable to severe COVID-19. The same effect, with similar magnitude, was observed for both primary series and booster vaccinations, suggesting a consistent underlying phenomenon, perhaps a lower likelihood of vaccination among seriously ill, end-of-life individuals, and less mobile elderly populations. COVID-19 booster vaccine policies should account for this effect when interpreting effectiveness estimates and formulating vaccine guidelines. Despite this effect, the results confirm strong protection from vaccination against severe forms of COVID-19.

## Contributors

HC co-designed the study, performed the statistical analyses, and co-wrote the first draft of the article. LJA conceived and co-designed the study, led the statistical analyses, and co-wrote the first draft of the article. HC and LJA accessed and verified all the data. PVC designed mass PCR testing to allow routine capture of variants and conducted viral genome sequencing. PT and MRH designed and conducted multiplex, RT-qPCR variant screening and viral genome sequencing. HMY and AAAT conducted viral genome sequencing. All authors contributed to data collection and acquisition, database development, discussion and interpretation of the results, and to the writing of the article. All authors have read and approved the final manuscript.

## Code availability

Standard epidemiological analyses were conducted using standard commands in Stata/SE 18.0.

## Acknowledgements

We acknowledge the many dedicated individuals at Hamad Medical Corporation, the Ministry of Public Health, the Primary Health Care Corporation, Qatar Biobank, Sidra Medicine, and Weill Cornell Medicine-Qatar for their diligent efforts and contributions to make this study possible. The authors are grateful for institutional salary support from the Biomedical Research Program and the Biostatistics, Epidemiology, and Biomathematics Research Core, both at Weill Cornell Medicine-Qatar, as well as for institutional salary support provided by the Ministry of Public Health, Hamad Medical Corporation, and Sidra Medicine. HC gratefully acknowledges salary support from the Junior Faculty Transition to Independence Program at Weill Cornell Medicine-Qatar and L'Oréal-UNESCO For Women In Science Middle East Regional Young Talents Program. The authors are also grateful for the Qatar Genome Programme and Qatar University Biomedical Research Center for institutional support for the reagents needed for the viral genome sequencing. Statements made herein are solely the responsibility of the authors.

## Additional information

### Competing interests

Adeel A Butt: has received institutional grant funding from Gilead Sciences unrelated to the work presented in this paper. The other authors declare that no competing interests exist.

### Funding

| Funder | Grant reference number | Author |
| --- | --- | --- |
| Weill Cornell Medicine-Qatar | Biomedical Research Program and the Biostatistics, Epidemiology, and Biomathematics Research Core | Hiam Chemaitelly |
| Weill Cornell Medicine-Qatar | Junior Faculty Transition to Independence Program | Hiam Chemaitelly |

| Funder | Grant reference number | Author |
|---|---|---|
| Ministry of Public Health | | Houssein H Ayoub<br>Peter Coyle<br>Patrick Tang<br>Mohammad R Hasan<br>Hadi M Yassine<br>Asmaa A Al Thani<br>Zaina Al Kanaani<br>Einas Al Kuwari<br>Andrew Jeremijenko<br>Anvar Hassan Kaleeckal<br>Ali Nizar Latif<br>Riyazuddin Mohammad Shaik<br>Hanan F Abdul Rahim<br>Gheyath K Nasrallah<br>Mohamed Ghaith Al Kuwari<br>Hamad Eid Al Romaihi<br>Mohamed H Al Thani<br>Abdullatif Al Khal<br>Roberto Bertollini<br>Adeel A Butt |
| Sidra Medicine | | Houssein H Ayoub<br>Peter Coyle<br>Patrick Tang<br>Mohammad R Hasan<br>Hadi M Yassine<br>Asmaa A Al Thani<br>Zaina Al Kanaani<br>Einas Al Kuwari<br>Andrew Jeremijenko<br>Anvar Hassan Kaleeckal<br>Ali Nizar Latif<br>Riyazuddin Mohammad Shaik<br>Hanan F Abdul Rahim<br>Gheyath K Nasrallah<br>Mohamed Ghaith Al Kuwari<br>Hamad Eid Al Romaihi<br>Mohamed H Al Thani<br>Abdullatif Al Khal<br>Roberto Bertollini<br>Adeel A Butt |
| Hamad Medical Corporation | | Houssein H Ayoub<br>Peter Coyle<br>Patrick Tang<br>Mohammad R Hasan<br>Hadi M Yassine<br>Asmaa A Al Thani<br>Zaina Al Kanaani<br>Einas Al Kuwari<br>Andrew Jeremijenko<br>Anvar Hassan Kaleeckal<br>Ali Nizar Latif<br>Riyazuddin Mohammad Shaik<br>Hanan F Abdul Rahim<br>Gheyath K Nasrallah<br>Mohamed Ghaith Al Kuwari<br>Hamad Eid Al Romaihi<br>Mohamed H Al Thani<br>Abdullatif Al Khal<br>Roberto Bertollini<br>Adeel A Butt |

| Funder | Grant reference number | Author |
|---|---|---|
| Qatar Genome Programme | | Houssein H Ayoub<br>Peter Coyle<br>Patrick Tang<br>Mohammad R Hasan<br>Hadi M Yassine<br>Asmaa A Al Thani<br>Zaina Al Kanaani<br>Einas Al Kuwari<br>Andrew Jeremijenko<br>Anvar Hassan Kaleeckal<br>Ali Nizar Latif<br>Riyazuddin Mohammad Shaik<br>Hanan F Abdul Rahim<br>Gheyath K Nasrallah<br>Mohamed Ghaith Al Kuwari<br>Hamad Eid Al Romaihi<br>Mohamed H Al Thani<br>Abdullatif Al Khal<br>Roberto Bertollini<br>Adeel A Butt |
| Qatar University Biomedical Research Center | | Houssein H Ayoub<br>Peter Coyle<br>Patrick Tang<br>Mohammad R Hasan<br>Hadi M Yassine<br>Asmaa A Al Thani<br>Zaina Al Kanaani<br>Einas Al Kuwari<br>Andrew Jeremijenko<br>Anvar Hassan Kaleeckal<br>Ali Nizar Latif<br>Riyazuddin Mohammad Shaik<br>Hanan F Abdul Rahim<br>Gheyath K Nasrallah<br>Mohamed Ghaith Al Kuwari<br>Hamad Eid Al Romaihi<br>Mohamed H Al Thani<br>Abdullatif Al Khal<br>Roberto Bertollini<br>Adeel A Butt |
| L'Oréal | For Women In Science Middle East Regional Young Talents Program | Hiam Chemaitelly |
| United Nations Educational, Scientific and Cultural Organization | For Women In Science Middle East Regional Young Talents Program | Hiam Chemaitelly |
| Qatar University | | Houssein H Ayoub<br>Peter Coyle<br>Hadi M Yassine<br>Asmaa A Al Thani<br>Gheyath K Nasrallah<br>Hanan F Abdul-Rahim<br>Laith J Abu-Raddad |

The funders had no role in study design, data collection and interpretation, or the decision to submit the work for publication.

## Author contributions

Hiam Chemaitelly, Resources, Data curation, Formal analysis, Validation, Investigation, Methodology, Writing – original draft, Writing – review and editing; Houssein H Ayoub, Zaina Al-Kanaani, Einas Al-Kuwari, Andrew Jeremijenko, Anvar Hassan Kaleeckal, Ali Nizar Latif, Riyazuddin Mohammad Shaik, Hanan F Abdul-Rahim, Gheyath K Nasrallah, Mohamed Ghaith Al-Kuwari, Hamad Eid Al-Romaihi,

Mohamed H Al-Thani, Abdullatif Al-Khal, Roberto Bertollini, Adeel A Butt, Resources, Data curation, Writing – review and editing; Peter Coyle, Patrick Tang, Mohammad R Hasan, Hadi M Yassine, Asmaa A Al Thani, Resources, Data curation, Investigation, Writing – review and editing; Laith J Abu-Raddad, Conceptualization, Resources, Data curation, Funding acquisition, Validation, Investigation, Methodology, Writing – original draft, Project administration, Writing – review and editing

### Author ORCIDs
Hiam Chemaitelly ⓘ https://orcid.org/0000-0002-8756-6968
Gheyath K Nasrallah ⓘ https://orcid.org/0000-0001-9252-1038
Laith J Abu-Raddad ⓘ https://orcid.org/0000-0003-0790-0506

### Ethics
The institutional review boards at Hamad Medical Corporation and Weill Cornell Medicine-Qatar approved this retrospective study with a waiver of informed consent.

### Decision letter and Author response
Decision letter https://doi.org/10.7554/eLife.103690.sa1
Author response https://doi.org/10.7554/eLife.103690.sa2

---

## Additional files

### Supplementary files
MDAR checklist

### Data availability
The dataset of this study is the property of the Qatar Ministry of Public Health and was provided to the researchers through a restricted-access agreement that prohibits sharing the dataset with third parties or making it publicly available. Access to the data is restricted to preserve the confidentiality of patient information and was granted to researchers for research purposes only. Individuals or entities interested in accessing the data may contact Dr. Hamad Al-Romaihi, Director of the Health Protection and Communicable Diseases Control Department at the Ministry of Public Health in Qatar, via email at halromaihi@MOPH.GOV.QA. All proposed research must obtain the necessary ethical approvals. Commercial use of the data is strictly prohibited. Requests for access are assessed by the Ministry of Public Health in Qatar, and approval is granted at its discretion. In compliance with data privacy laws and the data-sharing agreement with the Ministry of Public Health in Qatar, no datasets, whether raw or de-identified, can be publicly released by the researchers. However, aggregate data that do not compromise individual privacy are included within the manuscript and supplementary materials. This ensures transparency of the research findings and supports the reproducibility of results while maintaining compliance with legal requirements.

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

# Appendix 1

## Section S1: Study population and data sources

Qatar's national and universal public healthcare system uses the Cerner Millenium electronic medical record (EMR) system to track all the public healthcare encounters of each individual in the country, including all citizens and residents registered in the national and universal public healthcare system. Registration in the public healthcare system is mandatory for citizens and residents.

The databases analyzed in this study are data-extract downloads from the national EMR that have been implemented on a regular weekly schedule since the onset of pandemic by the Business Intelligence Unit at Hamad Medical Corporation (HMC). HMC is the national public healthcare provider in Qatar. At every download, all severe acute respiratory syndrome coronavirus 2 (SARS-CoV-2) tests, coronavirus disease 2019 (COVID-19) vaccinations, hospitalizations related to COVID-19, and all death records regardless of cause are provided to the authors through .csv files. These databases have been analyzed throughout the pandemic not only for study-related purposes, but also to provide policymakers with summary data and analytics to inform the national response.

Every health encounter in the national EMR is linked to an individual through the HMC Number, which serves as a unique identifier that links all records for this individual at the national level. Databases were merged and analyzed using the HMC Number to link all records pertaining to testing, vaccinations, hospitalizations, and deaths. All deaths in Qatar are recorded by the public healthcare system. COVID-19-related healthcare was provided exclusively in the public healthcare system. COVID-19 vaccination was also provided only through the public healthcare system. These health records were tracked throughout the COVID-19 pandemic using the national EMR system. This pre-established system ensured that we had access to comprehensive health records related to this study for both citizens and residents throughout the entire pandemic, allowing us to follow each person over time.

Demographic details for every HMC Number (individual) such as sex, age, and nationality are collected upon issuing of the universal health card, based on the Qatar Identity Card, which is a mandatory requirement by the Ministry of Interior to every citizen and resident in the country. Data extraction from the Qatar Identity Card to the digital health platform is performed electronically through scanning techniques.

All SARS-CoV-2 testing in any facility in Qatar is tracked nationally in one database, the national testing database. This database covers all testing throughout the country, whether in public or private facilities. Every polymerase chain reaction (PCR) test and a proportion of the facility-based rapid antigen tests conducted in Qatar, regardless of location or setting, are classified on the basis of symptoms and the reason for testing, such as the presence of clinical symptoms, contact tracing, participation in surveys or random testing campaigns, individual requests for testing, routine healthcare testing, pre-travel requirements, at the point of entry into the country, or any other relevant reasons for testing.

Before November 1, 2022, SARS-CoV-2 testing in Qatar was performed extensively with about 5% of the population tested every week (*Altarawneh et al., 2022b*). Based on the distribution of the reason for testing up to November 1, 2022, most of the tests in Qatar were conducted for routine reasons, such as travel-related purposes, and about 75% of infections were diagnosed not because of presence of symptoms (*Chemaitelly et al., 2021c*, *Altarawneh et al., 2022b*). Starting from November 1, 2022, testing for SARS-CoV-2 was substantially reduced, but still close to 1% of the population are being tested every week (*Chemaitelly et al., 2021c*). This study factored all SARS-CoV-2-related testing included in the national testing database over the duration of follow-up.

The first omicron wave that reached its peak in January of 2022 was massive and strained the testing capacity in the country (*Chemaitelly et al., 2023a*;, *Altarawneh et al., 2022b*, *Altarawneh et al., 2022c*, *Chemaitelly et al., 2022b*; *Chemaitelly et al., 2023b*). To alleviate the burden on PCR testing, rapid antigen testing was rapidly introduced. The swift change in testing policy precluded incorporating reason for testing for a number of rapid antigen tests. While the reason for testing is documented for all PCR tests, it is not uniformly available for all rapid antigen tests.

Rapid antigen test kits are accessible for purchase at pharmacies in Qatar, but results of home-based testing are neither reported nor documented in the national databases. Since SARS-CoV-2-test outcomes were linked to specific public health measures, restrictions, and privileges, testing policy and guidelines stress facility-based testing as the core testing mechanism in the population.

While facility-based testing is provided free of charge or at low subsidized costs, depending on the reason for testing, home-based rapid antigen testing is de-emphasized and not supported as part of national policy.

Qatar launched its COVID-19 vaccination program in December 2020, employing mRNA vaccines and prioritizing individuals based on coexisting conditions and age criteria (*Chemaitelly et al., 2021b*, *Abu-Raddad et al., 2022b*). COVID-19 vaccination was provided free of charge, regardless of citizenship or residency status, and was nationally tracked (*Chemaitelly et al., 2021b*, *Abu-Raddad et al., 2022b*).

Qatar has unusually young, diverse demographics, in that only 9% of its residents are ≥50 years of age, and 89% are expatriates from over 150 countries (*Planning and Statistics Authority-State of Qatar, 2020* , *Abu-Raddad et al., 2021a*). Further descriptions of the study population and these national databases were reported previously (*Abu-Raddad et al., 2021a*, *Chemaitelly et al., 2021c*, *Chemaitelly et al., 2021b*, *Chemaitelly et al., 2022a*, *Chemaitelly et al., 2022b*, *Chemaitelly et al., 2023b*, *Chemaitelly et al., 2023c*, *AlNuaimi et al., 2023*; *Abu-Raddad et al., 2022a*; *Altarawneh et al., 2022c*).

## Section S2: Laboratory methods and variant ascertainment

### Real-time reverse-transcription polymerase chain reaction testing

Nasopharyngeal and/or oropharyngeal swabs were collected for PCR testing and placed in Universal Transport Medium (UTM). Aliquots of UTM were: (1) extracted on KingFisher Flex (Thermo Fisher Scientific, USA), MGISP-960 (MGI, China), or ExiPrep 96 Lite (Bioneer, South Korea) followed by testing with real-time reverse-transcription PCR (RT-qPCR) using TaqPath COVID-19 Combo Kits (Thermo Fisher Scientific, USA) on an ABI 7500 FAST (Thermo Fisher Scientific, USA); (2) tested directly on the Cepheid GeneXpert system using the Xpert Xpress SARS-CoV-2 (Cepheid, USA); or (3) loaded directly into a Roche cobas 6800 system and assayed with the cobas SARS-CoV-2 Test (Roche, Switzerland). The first assay targets the viral S, N, and ORF1ab gene regions. The second targets the viral N and E-gene regions, and the third targets the ORF1ab and E-gene regions.

All PCR testing was conducted at the Hamad Medical Corporation Central Laboratory or Sidra Medicine Laboratory, following standardized protocols.

### Rapid antigen testing

SARS-CoV-2 antigen tests were performed on nasopharyngeal swabs using one of the following lateral flow antigen tests: Panbio COVID-19 Ag Rapid Test Device (Abbott, USA); SARS-CoV-2 Rapid Antigen Test (Roche, Switzerland); Standard Q COVID-19 Antigen Test (SD Biosensor, Korea); or CareStart COVID-19 Antigen Test (Access Bio, USA). All antigen tests were performed point-of-care according to each manufacturer's instructions at public or private hospitals and clinics throughout Qatar with prior authorization and training by the Ministry of Public Health (MOPH). Antigen test results were electronically reported to the MOPH in real time using the Antigen Test Management System which is integrated with the national Coronavirus Disease 2019 (COVID-19) database.

### Classification of infections by variant type

Surveillance for SARS-CoV-2 variants in Qatar is based on viral genome sequencing and multiplex RT-qPCR variant screening (*Vogels et al., 2021*) of weekly collected random positive clinical samples (*Abu-Raddad et al., 2021b* , *Chemaitelly et al., 2021c*, *National project of surveillance for variants of concern and viral genome sequencing, 2021*, *Benslimane et al., 2021*, *Hasan et al., 2021*, *Chemaitelly et al., 2021b*), complemented by deep sequencing of wastewater samples (*National project of surveillance for variants of concern and viral genome sequencing, 2021*, *Saththasivam et al., 2021*, *El-Malah et al., 2022*). Further details on the viral genome sequencing and multiplex RT-qPCR variant screening throughout the SARS-CoV-2 waves in Qatar can be found in previous publications (*National project of surveillance for variants of concern and viral genome sequencing, 2021*, *Abu-Raddad et al., 2021b*, *Chemaitelly et al., 2021c*, *Benslimane et al., 2021*, *Hasan et al., 2021*, *Chemaitelly et al., 2021b*, *Abu-Raddad et al., 2022a*, *Tang et al., 2021*, *Altarawneh et al., 2022b*, *Altarawneh et al., 2022c*, *Chemaitelly et al., 2022a*, *Qassim et al., 2022*, *Altarawneh et al., 2022a*, *Chemaitelly et al., 2023d*; *Chemaitelly et al., 2024*).

## Section S3: Classification of coexisting conditions

Coexisting conditions were ascertained and classified based on the ICD-10 codes for the conditions as recorded in the electronic health record encounters of each individual in the national EMR database that includes all citizens and residents registered in the national and universal public healthcare system. The public healthcare system provides healthcare to the entire resident population of Qatar free of charge or at heavily subsidized costs, including prescription drugs. With the mass expansion of this sector in recent years, facilities have been built to cater to specific needs of subpopulations. For example, tens of facilities have been built, including clinics and hospitals, in localities with high density of craft and manual workers (*Al Thani et al., 2021*).

All encounters for each individual were analyzed to determine the coexisting-condition classification for that individual. The national EMR database includes encounters starting from 2013, after this system was launched in Qatar. As long as each individual had at least one encounter with a specific coexisting-condition diagnosis since 2013, this person was classified with this coexisting condition. Individuals who may have coexisting conditions but never sought care in the public healthcare system were classified as individuals with no coexisting condition due to absence of recorded encounters for them.

The classification of coexisting conditions spanned the following conditions: (1) Behchet's disease, (2) cancer, (3) cardiovascular diseases, (4) infectious and parasitic diseases, (5) Chron's disease, (6) chronic kidney disease (CKD), (7) chronic liver disease (CLD), (8) chronic lung disease, (9) congenital malformations, deformations and chromosomal abnormalities, (10) diseases of the blood and blood-forming organs and certain disorders involving the immune mechanism, (11) diseases of the ear and mastoid process, (12) deep vein thrombosis (DVT), (13) dermatitis, (14) diabetes mellitus, (15) diseases of the circulatory system, (16) diseases of the digestive system, (17) diseases of the eye and adnex, (18) diseases of the genitourinary system, (19) diseases of the musculoskeletal system and connective tissue, (20) diseases of the nervous system, (21) diseases of the respiratory system, (22) diseases of the skin and subcutaneous tissue, (23) endocrine, nutritional and metabolic diseases, (24) gingivitis, (25) hypertension, (26) injury, poisoning and certain other consequences of external causes, (27) mental and behavioral disorders, (28) neoplasms, (29) periodontitis, (30) pregnancy, childbirth and the puerperium, (31) pulmonary tuberculosis, (32) rheumatoid arthritis, (33) Sjogren's syndrome, (34) stroke or neural conditions, (35) symptoms, signs and abnormal clinical and laboratory findings, not elsewhere classified, (36) systemic lupus erythematosus, (37) systemic sclerosis, (38) organ transplant, and (39) other unspecified factors influencing health status and contact with health services.

## Section S4: COVID-19 severity, criticality, and fatality classification

Classification of COVID-19 case severity (acute-care hospitalizations) (*World Health Organization, 2023b*), criticality (intensive-care-unit hospitalizations) (*World Health Organization, 2023b*), and fatality (*World Health Organization, 2023a*) followed World Health Organization (WHO) guidelines. Assessments were made by trained medical personnel independent of study investigators and using individual chart reviews, as part of a national protocol applied to every hospitalized COVID-19 patient. Each hospitalized COVID-19 patient underwent an infection severity assessment every three days until discharge or death. We classified individuals who progressed to severe, critical, or fatal COVID-19 between the time of the documented infection and the end of the study based on their worst outcome, starting with death (*World Health Organization, 2023a*), followed by critical disease (*World Health Organization, 2023b*), and then severe disease (*World Health Organization, 2023b*).

### Severe COVID-19

Severe COVID-19 disease was defined per WHO classification as a SARS-CoV-2 infected person with "oxygen saturation of <90% on room air, and/or respiratory rate of >30 breaths/minute in adults and children >5 years old (or ≥60 breaths/minute in children <2 months old or ≥50 breaths/minute in children 2-11 months old or ≥40 breaths/minute in children 1–5 years old), and/or signs of severe respiratory distress (accessory muscle use and inability to complete full sentences, and, in children, very severe chest wall indrawing, grunting, central cyanosis, or presence of any other general danger signs)" (*World Health Organization, 2023b*). Detailed WHO criteria for classifying Severe acute respiratory syndrome coronavirus 2 (SARS-CoV-2) infection severity can be found in the WHO technical report (*World Health Organization, 2023b*).

## Critical COVID-19

Critical COVID-19 disease was defined per WHO classification as a SARS-CoV-2 infected person with "acute respiratory distress syndrome, sepsis, septic shock, or other conditions that would normally require the provision of life sustaining therapies such as mechanical ventilation (invasive or non-invasive) or vasopressor therapy" (*World Health Organization, 2023b*). Detailed WHO criteria for classifying SARS-CoV-2 infection criticality can be found in the WHO technical report (*World Health Organization, 2023b*).

## Fatal COVID-19

COVID-19 death was defined per WHO classification as "a death resulting from a clinically compatible illness, in a probable or confirmed COVID-19 case, unless there is a clear alternative cause of death that cannot be related to COVID-19 disease (e.g. trauma). There should be no period of complete recovery from COVID-19 between illness and death. A death due to COVID-19 may not be attributed to another disease (e.g. cancer) and should be counted independently of preexisting conditions that are suspected of triggering a severe course of COVID-19". Detailed WHO criteria for classifying COVID-19 death can be found in the WHO technical report (*World Health Organization, 2023a*).

