## [Editor Report]

This fundamental work devoted to the effectiveness of COVID-19 vaccination is the first to calculate within a single paper the COVID vaccine effectiveness as well as a crucial confounder – the so-called healthy vaccinee effect/bias that influences results of observational vaccine effectiveness studies. Using rigorous methods and providing compelling evidence, the authors found a 65 % decrease in the likelihood of dying from non-COVID causes in the vaccinated individuals in the first six months after vaccination compared to the meticulously matched unvaccinated individuals. This indicates that observational studies on COVID-19 vaccines may inflate vaccine effectiveness, even if it is evaluated using the best available industry-standard methods. The work will be of broad interest not only to epidemiologists and vaccinologists but virtually to any scientist investigating the role of vaccines.

---

## [Decision Letter]

**Decision letter after peer review:**

Thank you for submitting your article "Assessing Healthy Vaccinee Effect in COVID-19 Vaccine Effectiveness Studies: A National Cohort Study in Qatar" for consideration by *eLife*. Your article has been reviewed by 3 peer reviewers, one of whom is a member of our Board of Reviewing Editors, and the evaluation has been overseen by Joshua Schiffer as the Senior Editor.

Essential Revisions:

1) Provide more details about the vaccine rollout strategy that was employed in Qatar and who was initially targeted for vaccination.

2) Provide sensitivity analysis removing criteria of negative test for controls for matching.

3) Provide sub-analysis of vaccine effectiveness against severe, critical, and fatal COVID in the first six months for comparison with the temporality of putative healthy vaccinee effect.

4) Provide further discussion of potential biases present in the study, and temper claims where warranted due to the high possibility of bias.

*Reviewer #1 (Recommendations for the authors):*

This study examines whether there is evidence of a healthy vaccinee effect, where those who opt for vaccination are on average healthier than those who remain unvaccinated, in Qatar in the context of COVID-19 vaccination. This question is interesting to assess because a healthy vaccinee effect would tend to bias estimates of eventual vaccine effectiveness, leading to potential underestimation of the vaccine's effectiveness since healthier individuals are more likely to become vaccinated. The study finds that the risk of non-COVID-19 related deaths was higher in vaccinated individuals, especially during the first 6 months after vaccination.

Particular strengths of this study include the high population coverage of the datasets, which include the entire population of Qatar and allow the linkage of all vaccinations and death records in the population, including non-permanent residents, as well as the very large sample size providing high statistical power for analyses.

I think perhaps the largest weakness of this study is the potential for selection bias. In particular, the decision to match vaccinated persons with unvaccinated persons who have a negative SARS-CoV-2 test could have led to selection bias because this means that different selection criteria were applied to vaccinated and unvaccinated individuals. Many of the reasons that would lead persons to get tested for SARS-CoV-2 would be related to health conditions and so might very plausibly be associated with their mortality risk, meaning that the selection of controls may not be independent from death risk. While the authors argue there is not much difference between a sensitivity analysis where they remove these matching criteria, I find there is a non-negligible difference in the hazard ratio estimates when these matching criteria are not applied, with the hazard ratios being somewhat attenuated. This suggests that while there is likely to be a healthy vaccinee effect, this effect may be slightly overestimated in the main analysis due to selection bias. I would have liked to see an analysis where these matching criteria were not applied in the full study population, not just in the subpopulation of Qatari citizens, who represent only some 9% of the total sample.

Another source of potential bias is that over half of all potentially eligible persons from the full cohort were not included in analyses because they could not be matched. Table 1 suggests that the excluded participants were older and had more comorbidities than those who were included in the matched cohort. It is therefore unclear whether the results of the analysis can be generalized to the entire population of Qatar given that over half were excluded.

While the analysis included the number of coexisting comorbidities in the matching process, I think it could be argued that comorbidities/pre-existing conditions are part of the healthy vaccinee effect that the authors are trying to capture, and so it might not be appropriate to match on this factor as comorbidities are part of the effect. Table 1 suggests that unvaccinated individuals are in fact less likely to have co-morbidities than vaccinated individuals. The healthy vaccinee effect might therefore have been overestimated by matching on this factor, as there is evidence that less healthy individuals were more likely to get vaccinated prior to matching.

While the above issues potentially could have led to bias of the results, I think the effect would still likely be present in analyses, so I would agree with the authors' main conclusion that there likely exists a healthy vaccinee effect. I also agree that it is however unclear to what extent this might have biased the estimates of vaccine effectiveness against COVID-19 related outcomes.

I thought the analysis of hazard ratios stratified by time since vaccination was particularly interesting and valuable, showing that the healthy vaccinee effect is mostly found in the first 6 months after vaccination. However, I would dispute the interpretation that the higher hazard ratios after 6 months represent confounding by indication (which the authors call the indication effect). It is likely that this is a bias caused by the depletion of susceptibles, where the unvaccinated individuals most likely to die are depleted at the start so the remaining unvaccinated survivors have a lower death rate, and is not because the vaccinated individuals were at higher risk of death (confounding by indication). I think the authors interpret this correctly in the discussion as they mention that this observation is potentially due to the depletion of seriously ill individuals in the unvaccinated, but I think they use the wrong terminology to refer to this effect, which would a bias due to depletion of susceptibles rather than confounding by indication.

– In general there could be more details about the vaccine rollout strategy that was employed in Qatar and who was initially targeted for vaccination, as this can help us understand who potentially may be overrepresented among the vaccinees. Most countries did not vaccinate at random, so if there were particular demographic criteria that were used to target vaccination this should be made clearer.

– I think that the criteria of a negative test for controls for matching could have led to selection bias as this criteria was not applied to vaccinated cases. I think there is enough of a difference in the sensitivity analyses when removing these criteria to suggest that this may have affected the estimates. I would have liked to see a sensitivity analysis for the full cohort, not just in Qataris, where either this criteria is dropped for the unvaccinated to assess the difference between hazard ratio estimates.

– It is not clear whether the cohort members who were excluded due to inability to match are systematically different from those who were included. I would have liked to see an estimate of the death probability in those excluded vs. included to assess the potential for selection bias.

– The authors interpret their hazard ratios as being evidence of a healthy vaccinee effect and indication effect in the results. In general it is better to reserve interpretation of the hazard ratios for the discussion, I would recommend simply reporting the hazard ratios in the results and reserving the interpretation of these as healthy vaccinee effect for the discussion.

– It is not clear based on the methods how the cause of death was assessed, as the authors indicate in the discussion they could not ascertain causes of death, so it is unclear how they know whether a death is due to COVID-19 or not.

– There is some discrepancy in the authors' interpretation of the hazard ratios >1, as they say, it is confounding by indication, but their more detailed explanation in the discussion discusses an effect that is due to depletion of susceptibles in the unvaccinated, not confounding due to indication for vaccination among healthy individuals. I think that the explanation of bias due to depletion of susceptibles in the unvaccinated is more plausible, as I would expect confounding by indication to be more present closer to vaccination (as it is health status at the time of vaccination which leads to confounding by indication, and this effect would be expected to diminish with time since vaccination). The authors should distinguish between these two hypotheses, as they are different.

– The authors may also want to consider a sensitivity analysis where they do not include comorbidities in the matching process because comorbidities are part of the healthy vaccinee effect that the authors are trying to measure; a person with less comorbidities is by definition healthier than a person with more comorbidities.

*Reviewer #2 (Recommendations for the authors):*

First of all, I would like to applaud the authors of this paper. Although several papers on HVE based on data from the Czech Republic were published in regard to COVID vaccines, this is by far the most thorough paper on healthy vaccinee effect in COVID-19 vaccines so far and the methods are meticulous. I particularly like the fact that the patients were matched also for prior COVID-19 infection – besides the reasons mentioned by the authors, another reason is that prior COVID-19 infection provides also excellent protection from a subsequent severe or critical COVID-19, which could otherwise skew their results. The paper shows also another important fact concerning HVE – this effect remains huge in the first six months despite a methodologically accurate and correct approach involving matching for a wide range of variables.

Still, I have several concerns that I feel need to be addressed, especially with statements that are, in my opinion, not entirely justified:

1) I disagree with the statement (Lines 71 and 395) that the effectiveness of the booster dose against severe, critical, or fatal COVID was 34.1% (95% CI: -46.4-76.7). In view of the wide and zero-containing confidence interval, this phrasing leads to misinterpretation. I would rather suggest rephrasing it as “No significant effect of the third dose of vaccine against severe, critical, or fatal COVID was demonstrated (34.1%; 95% CI: -46.4-76.7)”. After all, calculating any vaccine effectiveness from 6 vs 9 severe, critical, or fatal cases among approx. 70 thousand infections is doubtful. Rather, this result highlights a very good resistance to severe/critical/fatal COVID-19 in both these groups.

2) As rightly discussed by the Authors from Line 411 onwards, the healthy vaccinee bias is present at the beginning but after a couple of months (during which most of the frailest individuals die), HVE disappears, being replaced by indication bias. Hence, I am surprised that the authors did not perform a separate analysis of vaccine effectiveness against severe, critical, and fatal COVID in the first six months. In my opinion, such an analysis needs to be added to the paper, especially considering the fact that the vast majority of COVID deaths in Qatar within the study period occurred from March to June 2021, i.e., in the first 6 months after the majority of the elderly and frail individuals took their second vaccine dose. It is the same period in which the biggest healthy vaccinee effect (HVE) of 65 % (i.e., aHR for non-COVID deaths of 0.35) was observed – and comparison of the same period is necessary to correctly interpret the findings.

3) From the perspective of the approx. 65 % healthy vaccinee bias (i.e., aHR=0.35) in the first 6 months when most COVID-related deaths occurred, I think that the statement "Despite the presence of a healthy vaccinee effect, the results still confirm strong protection from vaccination against severe forms of COVID-19, as the observed effectiveness for the primary series was extremely high, at 96%" is too strong.

4) As mentioned in the summary of my review, it is, in my opinion, necessary to highlight the fact that HVE was present despite meticulous matching. This is actually the most important finding of this study – the failure to remove HVE through matching is crucial information for the whole bulk of observational studies on COVID-19 vaccination effectiveness. I suggest adding something along the lines that "Despite meticulous matching, large healthy vaccinee effect (aHR for non-COVID death in vaccinated vs unvaccinated of 0.35) was observed in the first six months of follow-up", probably to the Discussion as well as to the Abstract.

5) Seeing the virtually zero effect of booster vaccination, I would be very much interested in seeing the comparison of two doses vs the unvaccinated for the "after-booster" period (not saying that this is a must, I understand it is a lot of work; it could come as a new paper).

6) Calculating and reporting HVE over a period of several years is not optimal. The authors discuss this very well but still analyze the combined HVE+indication bias over the whole period and report it as one of the most prominent findings. I think that the individual periods (in which these two biases can be to a large degree separated) are much more interesting and important. It would be very helpful to calculate and compare vaccine effectiveness against severe, critical, or fatal COVID for each of the periods shown in Figure 2 and present it along with the results shown in Figure 2 now. This should be ideally also done separately for the most vulnerable group of 50+ years of age.

7) Was there any period with testing bias in Qatar? In other words, was there any period when the vaccinated did not have to test for COVID-19 while the unvaccinated had to test? If so, could it have influenced the results of your study? I think the paper would benefit from this discussion.

8) I think that the peculiar composition of Qatar's population is leading to problems with low numbers, especially in the elderly population most vulnerable to COVID-19, is basically the only limitation of this study. However, the reported pattern is in line with what was observed elsewhere and the authors did, as mentioned above, an excellent job in making up for this limitation by meticulous matching so I consider their work highly relevant and well performed.

*Reviewer #3 (Recommendations for the authors):*

The authors have conducted cohort studies using routinely collected data from Qatar. These datasets have been the basis of a series of high-quality and influential studies that were conducted in Qatar into COVID-19 vaccine effectiveness (VE) during the pandemic.

In this paper the authors are less focused on VE and more on defining the degree and temporal profile of a possible early 'healthy vaccinee effect' followed by an 'indication bias'. The category of 'non-COVID' death is a potentially useful negative outcome control. Deviation from the null indicates bias if the vaccines truly have no effect on the study outcomes. However, its value here is reduced by the lack of information on cause of death. In this study it may be a reasonable directional bias indicator but does not specify the cause of bias. I believe their methods and analyses are robust and they are observing real effects. However, the authors do not present direct data on causes of death or variables that would help define healthy vaccinees or underpin the claim of indication bias. I think they need to provide a more comprehensive defence of their conclusions.

1). It would be helpful if the authors defined the boundaries of the 'healthy vaccinee effect' they are investigating with their data. The closest to a definition I could find was in the Introduction where the authors state 'the healthy vaccinee effect occurs when healthier or health-conscious individuals are more likely to receive vaccination'. In this setting 'health-conscious individuals' are making a deliberate choice to be vaccinated, and this is consistent with other choices they make (exercise, avoiding smoking and excess alcohol, eating a healthy diet, undergoing screening). At times in the paper the authors appear to equate the healthy vaccinee effect with the absence of underlying disease and, at its most extreme, the absence of disease that would lead to avoidance of vaccination because of clinical futility. There is no doubt that omission of vaccination is practiced in patients with terminal disease, but I think it should be regarded as a separate bias and its absence does not denote a 'healthy vaccinee effect'. I think this needs clarification.

2) I think the overall relationship between comorbidity and frailty and vaccination choice needs deeper discussion. The authors cite community studies of influenza vaccination, which generally have lower population coverage than COVD-19 vaccination, allowing more room for selection biases. While older independent community members who are active may demonstrate a greater preference for influenza vaccination, this will not be the case for those in long-term care facilities, where a common policy is to vaccinate all residents.

3) The authors also mention confounding by indication, which with COVID-19 has been influenced by government policies to provide early vaccination to high-risk groups and healthcare workers. As they say, this runs counter to the presumed healthy vaccinee effect. These policies change quickly and can introduce time-varying exposures in certain groups. I am uncertain if vaccination policies were considered as potential sources of bias – but they may be controlled by the decision to match on calendar week. The same comment applies to 'environmental risk' – variation in community attack rates over time and region.

4) The authors seem somewhat dismissive of misclassification of deaths as contributing to their findings. As noted, they don't have comprehensive cause of death data. I agree that a non-specific immunostimulant benefit of the vaccines is unlikely. However, some COVID-related deaths may be missed during coding. For instance, someone who dies of a heart attack or stroke 4 weeks after a serious COVID illness. The coding decision could also be influenced by knowledge of vaccination status as it appears that coders were unblinded.

5) But misclassification of outcomes is not the only potential source of bias. Because the authors don't have direct evidence to support healthy user bias and confounding by indication, I think they should use a bias framework (e.g., ROBINS-I) to discuss and (as appropriate) reject the other potential causes of bias

6) I am not arguing their assertions are wrong. But I think their language is over-confident and they need to make a more comprehensive case to back their conclusions.

7) This is an accomplished group performing sophisticated studies. However, the level of self-citation seems excessive. In some cases, it is justified to anchor the current work. But there is a large literature on the effects of COVID-19 on COVID mortality with a significant number of studies that also reported all causes of non-COVID mortality. This is not reflected in the authors' choice of references.

---

## [Author Response]

Essential Revisions:1) Provide more details about the vaccine rollout strategy that was employed in Qatar and who was initially targeted for vaccination.

A subsection has been added to describe the rollout strategy employed in Qatar, including the initial target groups prioritized for vaccination (Methods, Page 8, Paragraph 3 and Page 9, Paragraphs 1-2). Additionally, a figure has been included to illustrate the rollout of both the primary vaccination series and booster doses (Figure 1—figure supplement 1).

In this context, we would also like to highlight a key strength of this study. While vaccination policies evolve over time, potentially introducing time-varying exposures across different groups, we addressed this by matching participants by calendar week and initiating follow-up from this time point. This approach ensured that each matched pair was recruited into the cohorts during a period with similar vaccination policies and practices. This point has now been clarified in the revised manuscript (Methods, Page 12, Paragraph 3).

2) Provide sensitivity analysis removing criteria of negative test for controls for matching.

This sensitivity analysis was already included in the original manuscript but was restricted to the cohort of Qatari nationals. This subgroup represents the segment of the population that can be confidently assumed to have continuous presence in Qatar. Conducting this analysis for the entire population, as suggested by the first reviewer, may introduce bias, as it removes the requirement for evidence of presence in Qatar.

The expatriate population in Qatar primarily consists of craft and manual workers, and this demographic has experienced turnover due to infrastructure projects associated with the 2022 FIFA World Cup. Consequently, comparing vaccinated individuals, whose presence in Qatar is verified through their vaccination records, with unvaccinated individuals, for whom there is uncertainty about presence and timing, risks underestimating death and infection incidence among the unvaccinated group. Including a sensitivity analysis under such conditions, where there is a reasonable basis to suspect bias, could compromise the validity of the results, and we were therefore hesitant to include such an analysis.

To address this request and the reviewer's concern, we conducted the suggested sensitivity analysis, and the results are presented in Author response image 1. The analysis reaffirmed, with statistical significance, the same observed healthy vaccinee effect and its characteristic temporal evolution as in the main analysis. However, as expected, the effect size was smaller due to bias introduced by the under ascertainment of deaths in the unvaccinated group.

Given these considerations, we are inclined to retain the original sensitivity analysis restricted to Qataris and exclude the new analysis for the entire population due to concerns about bias affecting the accuracy of the effect size estimation, even though the new analysis qualitatively demonstrated similar results.

**Author response image 1. sa2fig1:** Sensitivity analysis. Adjusted hazard ratios for incidence of non-COVID-19 death in the entire population without matching on a SARS-CoV-2-negative test among controls. Results are shown for the (A) two-dose analysis and the (B) three-dose analysis, in the first 6 months of follow-up and the period thereafter.

3) Provide sub-analysis of vaccine effectiveness against severe, critical, and fatal COVID in the first six months for comparison with the temporality of putative healthy vaccinee effect.

Excellent idea, thank you. This sub analysis has now been incorporated, examining overall effectiveness as well as effectiveness within the subgroups of individuals aged <50 years and those aged ≥50 years, as also requested by Reviewer 2 (Figure 2—figure supplement 1). The findings were consistent with the main study results. This addition has been also described in the Results section (Results, Page 19, Paragraph 1 and Page 20, Paragraph 4).

4) Provide further discussion of potential biases present in the study, and temper claims where warranted due to the high possibility of bias.

The manuscript has been substantially revised and enhanced with a more detailed discussion of potential biases, as well as additional analyses, all of which support the study findings (changes throughout the manuscript). Claims have been tempered where appropriate to ensure a balanced interpretation of the results. For a comprehensive overview of these changes, please note our detailed responses to the reviewers' comments, as outlined below.

Reviewer #1 (Recommendations for the authors):This study examines whether there is evidence of a healthy vaccinee effect, where those who opt for vaccination are on average healthier than those who remain unvaccinated, in Qatar in the context of COVID-19 vaccination. This question is interesting to assess because a healthy vaccinee effect would tend to bias estimates of eventual vaccine effectiveness, leading to potential underestimation of the vaccine's effectiveness since healthier individuals are more likely to become vaccinated. The study finds that the risk of non-COVID-19 related deaths was higher in vaccinated individuals, especially during the first 6 months after vaccination.Particular strengths of this study include the high population coverage of the datasets, which include the entire population of Qatar and allow the linkage of all vaccinations and death records in the population, including non-permanent residents, as well as the very large sample size providing high statistical power for analyses.I think perhaps the largest weakness of this study is the potential for selection bias. In particular, the decision to match vaccinated persons with unvaccinated persons who have a negative SARS-CoV-2 test could have led to selection bias because this means that different selection criteria were applied to vaccinated and unvaccinated individuals. Many of the reasons that would lead persons to get tested for SARS-CoV-2 would be related to health conditions and so might very plausibly be associated with their mortality risk, meaning that the selection of controls may not be independent from death risk. While the authors argue there is not much difference between a sensitivity analysis where they remove these matching criteria, I find there is a non-negligible difference in the hazard ratio estimates when these matching criteria are not applied, with the hazard ratios being somewhat attenuated. This suggests that while there is likely to be a healthy vaccinee effect, this effect may be slightly overestimated in the main analysis due to selection bias. I would have liked to see an analysis where these matching criteria were not applied in the full study population, not just in the subpopulation of Qatari citizens, who represent only some 9% of the total sample.

We appreciate the reviewer's perspective, but we respectfully disagree with the assertion that there is a non-negligible difference in the hazard ratio estimates in the cited sensitivity analyses when the matching criteria are not applied. The hazard ratios were 0.29 (95% CI: 0.19–0.43) and 0.38 (95% CI: 0.30–0.50), both with overlapping 95% confidence intervals.

It is important to note that these two hazard ratios were derived from analyses conducted on two distinct cohort studies, each involving different cohorts, as the entire matching process had to be repeated for the analysis in which the matching to a negative SARS-CoV-2 test was removed. Consequently, due to sampling variation, the hazard ratios were not identical; however, they remain consistent within overlapping 95% confidence intervals. Even if the exact same analysis were repeated under identical conditions, the hazard ratios would, as the reviewer would appreciate, still vary due to sampling variation, but the estimates would fall within overlapping 95% confidence intervals.

We also appreciate the reviewer's point that, under normal circumstances, many of the factors prompting individuals to undergo SARS-CoV-2 testing might be related to underlying health conditions and could plausibly be associated with mortality risk. However, this association is much less likely in the context of the COVID-19 pandemic due to the extensive testing conducted primarily for routine reasons unrelated to health conditions. Until October 31, 2022, Qatar implemented a widespread testing strategy, testing 5% of the population weekly, primarily for routine purposes such as screening or travel-related requirements [1, 2].

We conducted the sensitivity analysis removing the matching requirement for a SARS-CoV-2-negative test exclusively for Qataris, as this subgroup represents the portion of the population that can be confidently assumed to be continuously present in Qatar. Conducting this analysis for the entire population, as suggested by the reviewer, may introduce bias, as it removes the requirement for evidence of presence in Qatar.

The expatriate population in Qatar primarily consists of craft and manual workers, and this demographic has experienced turnover due to infrastructure projects associated with the 2022 FIFA World Cup. Consequently, comparing vaccinated individuals, whose presence in Qatar is verified through their vaccination records, with unvaccinated individuals, for whom there is uncertainty about presence and timing, risks underestimating death and infection incidence among the unvaccinated group. Including a sensitivity analysis under such conditions, where there is a reasonable basis to suspect bias, could compromise the validity of the results, and we were therefore hesitant to include such an analysis.

To address the reviewer's concern, we have now conducted the suggested sensitivity analysis, and the results are presented in Author response image 1. The analysis reaffirmed, with statistical significance, the same observed healthy vaccinee effect and its characteristic temporal evolution as in the main analysis. However, as expected, the effect size was smaller due to bias introduced by the under ascertainment of deaths in the unvaccinated group.

Given these considerations, we are inclined to retain the original sensitivity analysis restricted to Qataris and exclude the new analysis for the entire population due to concerns about bias affecting the accuracy of the effect size estimation, even though the new analysis qualitatively demonstrated similar results.

In this context, we would also like to indicate a key strength of our matching approach. As vaccination policies evolved over time, potentially introducing time-varying exposures across different groups, we addressed this issue by matching participants by calendar week (using the matching by negative SARS-CoV-2 test) and initiating follow-up from this time point. This approach ensured that each matched pair was recruited into the cohorts during a period with similar vaccination policies and practices. This point has now been clarified in the revised manuscript (Methods, Page 12, Paragraph 3).

Another source of potential bias is that over half of all potentially eligible persons from the full cohort were not included in analyses because they could not be matched. Table 1 suggests that the excluded participants were older and had more comorbidities than those who were included in the matched cohort. It is therefore unclear whether the results of the analysis can be generalized to the entire population of Qatar given that over half were excluded.

The study was designed to address a specific research question: the existence of the healthy vaccinee effect in rigorously conducted vaccine effectiveness studies. As such, the focus is on the internal validity of the study, that is ensuring that the relationship between the exposure and outcome is measured accurately and free from bias and confounding. This approach is analogous to the design of vaccine efficacy RCTs, which are typically conducted in select populations that may not fully represent the wider population (e.g., excluding pregnant women, children, or individuals with specific coexisting conditions). The purpose of such RCTs, and similarly this study, is not to provide population-wide estimates but to generate a precise and unbiased measure of the exposure-outcome relationship that is being investigated.

To achieve this goal, the study employed rigorous matching across multiple factors to control for selection bias, similar to the role of randomization in an RCT. While this matching process resulted in study cohorts that are not fully representative of Qatar's population, this was intentional and aligns with the study's objective. The aim was not to estimate the healthy vaccinee effect for Qatar's broader population but to produce a valid and accurate measure of this specific effect under investigation and its existence.

This point has now been discussed and clarified in the revised manuscript (Discussion, Page 25, Paragraphs 3-4).

While the analysis included the number of coexisting comorbidities in the matching process, I think it could be argued that comorbidities/pre-existing conditions are part of the healthy vaccinee effect that the authors are trying to capture, and so it might not be appropriate to match on this factor as comorbidities are part of the effect. Table 1 suggests that unvaccinated individuals are in fact less likely to have co-morbidities than vaccinated individuals. The healthy vaccinee effect might therefore have been overestimated by matching on this factor, as there is evidence that less healthy individuals were more likely to get vaccinated prior to matching.

We appreciate the reviewer's point but respectfully disagree. The objective of this study is not to describe differences between individuals who receive the vaccine and those who do not but to assess the healthy vaccinee effect in specifically rigorously conducted vaccine effectiveness studies. In such studies, it is critical to control for observable differences in health status to ensure an unbiased measure of vaccine effectiveness. Rigorous vaccine effectiveness studies have conventionally addressed this by controlling for coexisting conditions using administrative healthcare utilization databases.

Please note our detailed discussion of this point in the Introduction (Introduction, Page 5, Paragraph 3 and Page 6, Paragraphs 1-2), as well as the expanded Methods (Methods Page 9, Paragraph 4 and Page 10, Paragraph 1). To further clarify the study's objective and avoid potential confusion, we have also emphasized this point in the Abstract and at the opening of the Discussion section (Abstract and Discussion, Page 20, Paragraph 5).

While the above issues potentially could have led to bias of the results, I think the effect would still likely be present in analyses, so I would agree with the authors' main conclusion that there likely exists a healthy vaccinee effect. I also agree that it is however unclear to what extent this might have biased the estimates of vaccine effectiveness against COVID-19 related outcomes.

The manuscript has been substantially revised and enhanced to include a more detailed discussion of potential biases, as well as additional analyses, all of which support the study findings (changes incorporated throughout the manuscript). We hope these revisions address the reviewers' concerns.

I thought the analysis of hazard ratios stratified by time since vaccination was particularly interesting and valuable, showing that the healthy vaccinee effect is mostly found in the first 6 months after vaccination. However, I would dispute the interpretation that the higher hazard ratios after 6 months represent confounding by indication (which the authors call the indication effect). It is likely that this is a bias caused by the depletion of susceptibles, where the unvaccinated individuals most likely to die are depleted at the start so the remaining unvaccinated survivors have a lower death rate, and is not because the vaccinated individuals were at higher risk of death (confounding by indication). I think the authors interpret this correctly in the discussion as they mention that this observation is potentially due to the depletion of seriously ill individuals in the unvaccinated, but I think they use the wrong terminology to refer to this effect, which would a bias due to depletion of susceptibles rather than confounding by indication.

We appreciate the reviewer's feedback on this analysis and for raising this point. We also apologize for the confusion caused by these statements. This part of the discussion aimed to address a salient point in the findings. It is possible that both a healthy vaccinee effect and an indication effect could coexist, though they likely operate at different strengths and timeframes. The healthy vaccinee effect is primarily driven by seriously ill and end-of-life individuals, such as terminal cancer patients, as well as frail and less mobile elderly persons. In contrast, the indication effect would be driven by individuals with less immediately life-threatening but still serious health conditions who would pursue vaccination to reduce the risk of complicating their health conditions with infection, such as heart disease or diabetes, which may increase mortality risk over a longer time horizon rather than immediately.

Our results indicate that the healthy vaccinee effect was pronounced during the first six months after vaccination. However, over the longer term, there was an increased risk of mortality among those vaccinated. While this is primarily attributed to the depletion of seriously ill individuals in the unvaccinated group during the initial six months, leaving a relatively healthier cohort, it is also not inconsistent with a potential presence of some indication effect. This effect would be driven by individuals with conditions such as heart disease or diabetes, who sought vaccination to reduce the risk of infection exacerbating their health conditions but who inherently have a higher mortality risk over time.

To address this comment, we have now substantially revised this part of the discussion to address the reviewer's point (Discussion, Page 21, Paragraph 4 and Page 22, Paragraph 1). Additionally, we have removed references to indication bias in contexts where it might cause confusion (Results, Page 18, Paragraph 2 and Page 20, Paragraph 1).

– In general there could be more details about the vaccine rollout strategy that was employed in Qatar and who was initially targeted for vaccination, as this can help us understand who potentially may be overrepresented among the vaccinees. Most countries did not vaccinate at random, so if there were particular demographic criteria that were used to target vaccination this should be made clearer.

Thank you for the useful suggestion. A subsection has been added to describe the rollout strategy employed in Qatar, including the initial target groups prioritized for vaccination (Methods, Page 8, Paragraph 3 and Page 9, Paragraphs 1-2). Additionally, a figure has been included to illustrate the rollout of both the primary vaccination series and booster doses (Figure 1—figure supplement 1).

In this context, we would also like to indicate a key strength of our matching approach. As vaccination policies evolved over time, potentially introducing time-varying exposures across different groups, we addressed this issue by matching participants by calendar week (using the matching by negative SARS-CoV-2 test) and initiating follow-up from this time point. This approach ensured that each matched pair was recruited into the cohorts during a period with similar vaccination policies and practices. This point has now been clarified in the revised manuscript (Methods, Page 12, Paragraph 3).

– I think that the criteria of a negative test for controls for matching could have led to selection bias as this criteria was not applied to vaccinated cases. I think there is enough of a difference in the sensitivity analyses when removing these criteria to suggest that this may have affected the estimates. I would have liked to see a sensitivity analysis for the full cohort, not just in Qataris, where either this criteria is dropped for the unvaccinated to assess the difference between hazard ratio estimates.

We appreciate the reviewer's perspective, but we respectfully disagree with the assertion that there is a non-negligible difference in the hazard ratio estimates in the cited sensitivity analyses when the matching criteria are not applied. The hazard ratios were 0.29 (95% CI: 0.19–0.43) and 0.38 (95% CI: 0.30–0.50), both with overlapping 95% confidence intervals.

It is important to note that these two hazard ratios were derived from analyses conducted on two distinct cohort studies, each involving different cohorts, as the entire matching process had to be repeated for the analysis in which the matching to a negative SARS-CoV-2 test was removed. Consequently, due to sampling variation, the hazard ratios were not identical; however, they remain consistent within overlapping 95% confidence intervals. Even if the exact same analysis were repeated under identical conditions, the hazard ratios would still vary due to sampling variation, but the estimates would fall within overlapping 95% confidence intervals.

We also appreciate the reviewer's point that, under normal circumstances, many of the factors prompting individuals to undergo SARS-CoV-2 testing might be related to underlying health conditions and could plausibly be associated with mortality risk. However, this association is much less likely in the context of the COVID-19 pandemic due to the extensive testing conducted primarily for routine reasons unrelated to health conditions. Until October 31, 2022, Qatar implemented a widespread testing strategy, testing 5% of the population weekly, primarily for routine purposes such as screening or travel-related requirements [1, 2].

We conducted the sensitivity analysis removing the matching requirement for a SARS-CoV-2-negative test exclusively for Qataris, as this subgroup represents the portion of the population that can be confidently assumed to be continuously present in Qatar. Conducting this analysis for the entire population, as suggested by the reviewer, may introduce bias, as it removes the requirement for evidence of presence in Qatar.

The expatriate population in Qatar primarily consists of craft and manual workers, and this demographic has experienced turnover due to infrastructure projects associated with the 2022 FIFA World Cup. Consequently, comparing vaccinated individuals, whose presence in Qatar is verified through their vaccination records, with unvaccinated individuals, for whom there is uncertainty about presence and timing, risks underestimating death and infection incidence among the unvaccinated group. Including a sensitivity analysis under such conditions, where there is a reasonable basis to suspect bias, could compromise the validity of the results, and we were therefore hesitant to include such an analysis.

To address the reviewer's concern, we have now conducted the suggested sensitivity analysis, and the results are presented in Author response image 1. The analysis reaffirmed, with statistical significance, the same observed healthy vaccinee effect and its characteristic temporal evolution as in the main analysis. However, as expected, the effect size was smaller due to bias introduced by the under ascertainment of deaths in the unvaccinated group.

Given these considerations, we are inclined to retain the original sensitivity analysis restricted to Qataris and exclude the new analysis for the entire population due to concerns about bias affecting the accuracy of the effect size estimation, even though the new analysis qualitatively demonstrated similar results.

– It is not clear whether the cohort members who were excluded due to inability to match are systematically different from those who were included. I would have liked to see an estimate of the death probability in those excluded vs. included to assess the potential for selection bias.

The study was designed to address a specific research question: the existence of the healthy vaccinee effect in rigorously conducted vaccine effectiveness studies. As such, the focus is on the internal validity of the study, that is ensuring that the relationship between the exposure and outcome is measured accurately and free from bias and confounding. This approach is analogous to the design of vaccine efficacy RCTs, which are typically conducted in select populations that may not fully represent the wider population (e.g., excluding pregnant women, children, or individuals with specific coexisting conditions). The purpose of such RCTs, and similarly this study, is not to provide population-wide estimates but to generate a precise and unbiased measure of the exposure-outcome relationship that is being investigated. Therefore, differences between matched and unmatched groups are not of direct consequence for the specific research question addressed in this study.

To address the specific research question of this study, the study employed rigorous matching across multiple factors to control for selection bias, similar to the role of randomization in an RCT. While this matching process resulted in study cohorts that are not fully representative of Qatar's population, this was intentional and aligns with the study's objective. The aim was not to estimate the healthy vaccinee effect for Qatar's broader population but to produce a valid and accurate measure of the exposure-outcome relationship, specifically the effect under investigation and its existence.

This point has now been discussed and clarified in the revised manuscript (Discussion, Page 25, Paragraphs 3-4).

– The authors interpret their hazard ratios as being evidence of a healthy vaccinee effect and indication effect in the results. In general it is better to reserve interpretation of the hazard ratios for the discussion, I would recommend simply reporting the hazard ratios in the results and reserving the interpretation of these as healthy vaccinee effect for the discussion.

We fully appreciate the reviewer's point, and during the original drafting of this manuscript, the reviewer's suggested approach was initially considered as our plan. However, we found that it disrupted the flow of the narrative and arguments, leading to potential for confusion. Given the multiple subtle and interconnected concepts in this study, we concluded that the most effective approach was to provide only the immediate interpretation (but not the implications) of the hazard ratios at the point of presentation. This approach ensures greater clarity and coherence in conveying the findings. Additionally, we have now removed all references to indication bias in the Results section (Results, Page 18, Paragraph 2 and Page 20, Paragraph 1).

– It is not clear based on the methods how the cause of death was assessed, as the authors indicate in the discussion they could not ascertain causes of death, so it is unclear how they know whether a death is due to COVID-19 or not.

COVID-19 deaths in Qatar were systematically identified through a national protocol applied to every COVID-19 case. However, this protocol was designed to determine whether a death was classified as a COVID-19 death or not, and did not investigate the specific detailed causes of non-COVID-19 deaths (Discussion, Page 23, Paragraph 3).

To clarify this matter, we have now expanded the description of this aspect of the methods into a subsection, detailing the classification process for COVID-19 deaths (Methods, Page 10, Paragraphs 3-4 and Page 11, Paragraphs 1-3).

– There is some discrepancy in the authors' interpretation of the hazard ratios >1, as they say, it is confounding by indication, but their more detailed explanation in the discussion discusses an effect that is due to depletion of susceptibles in the unvaccinated, not confounding due to indication for vaccination among healthy individuals. I think that the explanation of bias due to depletion of susceptibles in the unvaccinated is more plausible, as I would expect confounding by indication to be more present closer to vaccination (as it is health status at the time of vaccination which leads to confounding by indication, and this effect would be expected to diminish with time since vaccination). The authors should distinguish between these two hypotheses, as they are different.

We agree with the reviewer that it was necessary to clearly distinguish between these two hypotheses. This part of the discussion aimed to address a salient point in the findings. It is possible that both a healthy vaccinee effect and an indication effect could coexist, though they likely operate at different strengths and timeframes. The healthy vaccinee effect is primarily driven by seriously ill and end-of-life individuals, such as terminal cancer patients, as well as frail and less mobile elderly persons. In contrast, the indication effect would be driven by individuals with less immediately life-threatening but still serious health conditions who would pursue vaccination to reduce the risk of complicating their health conditions with infection, such as heart disease or diabetes, which may increase mortality risk over a longer time horizon rather than immediately.

Our results indicate that the healthy vaccinee effect was pronounced during the first six months after vaccination. However, over the longer term, there was an increased risk of mortality among those vaccinated. While this is primarily attributed to the depletion of seriously ill individuals in the unvaccinated group during the initial six months, leaving a relatively healthier cohort, it is also not inconsistent with a potential presence of some indication effect. This effect would be driven by individuals with conditions such as heart disease or diabetes, who sought vaccination to reduce the risk of infection exacerbating their health conditions but who inherently have a higher mortality risk over time.

To clarify these nuances and distinguish between these two hypotheses, we have now substantially revised this part of the discussion to address the reviewer's point (Discussion, Page 21, Paragraph 4 and Page 22, Paragraph 1). Additionally, we have removed references to indication bias in contexts where it might cause confusion (Results, Page 18, Paragraph 2 and Page 20, Paragraph 1).

– The authors may also want to consider a sensitivity analysis where they do not include comorbidities in the matching process because comorbidities are part of the healthy vaccinee effect that the authors are trying to measure; a person with less comorbidities is by definition healthier than a person with more comorbidities.

We appreciate the reviewer's comment but respectfully do not agree with this point. The objective of this study is not to describe differences between individuals who receive the vaccine and those who do not but to assess the healthy vaccinee effect in specifically rigorously conducted vaccine effectiveness studies. In such studies, it is critical to control for observable differences in health status to ensure an unbiased measure of vaccine effectiveness. Rigorous vaccine effectiveness studies have conventionally addressed this by controlling for coexisting conditions using administrative healthcare utilization databases.

Please note our detailed discussion of this point in the Introduction (Introduction, Page 5, Paragraph 3 and Page 6, Paragraphs 1-2), as well as the expanded Methods (Methods Page 9, Paragraph 4 and Page 10, Paragraph 1). To further clarify the study's objective and avoid potential confusion, we have also emphasized this point in the Abstract and at the opening of the Discussion section (Abstract and Discussion, Page 20, Paragraph 5).

Reviewer #2 (Recommendations for the authors):First of all, I would like to applaud the authors of this paper. Although several papers on HVE based on data from the Czech Republic were published in regard to COVID vaccines, this is by far the most thorough paper on healthy vaccinee effect in COVID-19 vaccines so far and the methods are meticulous. I particularly like the fact that the patients were matched also for prior COVID-19 infection – besides the reasons mentioned by the authors, another reason is that prior COVID-19 infection provides also excellent protection from a subsequent severe or critical COVID-19, which could otherwise skew their results. The paper shows also another important fact concerning HVE – this effect remains huge in the first six months despite a methodologically accurate and correct approach involving matching for a wide range of variables.

We thank the reviewer for the time and effort put into this review, the assessment of our work, and the constructive feedback on our manuscript that enriched it and improved its readability. Please find below a point-by-point reply addressing each of the reviewer's comments.

We thank the reviewer for raising the useful point regarding the matching by prior infection status, which has now been indicated in the revised manuscript (Methods, Page 12, Paragraph 2).

Still, I have several concerns that I feel need to be addressed, especially with statements that are, in my opinion, not entirely justified:1) I disagree with the statement (Lines 71 and 395) that the effectiveness of the booster dose against severe, critical, or fatal COVID was 34.1% (95% CI: -46.4-76.7). In view of the wide and zero-containing confidence interval, this phrasing leads to misinterpretation. I would rather suggest rephrasing it as „No significant effect of the third dose of vaccine against severe, critical, or fatal COVID was demonstrated (34.1%; 95% CI: -46.4-76.7). After all, calculating any vaccine effectiveness from 6 vs 9 severe, critical, or fatal cases among approx. 70 thousand infections is doubtful. Rather, this result highlights a very good resistance to severe/critical/fatal COVID-19 in both these groups.

We agree with the reviewer. This statement has now been edited as suggested (Results, Page 20, Paragraph 4).

2) As rightly discussed by the Authors from Line 411 onwards, the healthy vaccinee bias is present at the beginning but after a couple of months (during which most of the frailest individuals die), HVE disappears, being replaced by indication bias. Hence, I am surprised that the authors did not perform a separate analysis of vaccine effectiveness against severe, critical, and fatal COVID in the first six months. In my opinion, such an analysis needs to be added to the paper, especially considering the fact that the vast majority of COVID deaths in Qatar within the study period occurred from March to June 2021, i.e., in the first 6 months after the majority of the elderly and frail individuals took their second vaccine dose. It is the same period in which the biggest healthy vaccinee effect (HVE) of 65 % (i.e., aHR for non-COVID deaths of 0.35) was observed – and comparison of the same period is necessary to correctly interpret the findings.

Excellent idea, thank you. This sub analysis has now been incorporated, examining overall effectiveness as well as effectiveness within the subgroups of individuals aged <50 years and those aged ≥50 years (Figure 2—figure supplement 1). This addition has been also described in the Results section (Results, Page 19, Paragraph 1 and Page 20, Paragraph 4).

3) From the perspective of the approx. 65 % healthy vaccinee bias (i.e., aHR=0.35) in the first 6 months when most COVID-related deaths occurred, I think that the statement "Despite the presence of a healthy vaccinee effect, the results still confirm strong protection from vaccination against severe forms of COVID-19, as the observed effectiveness for the primary series was extremely high, at 96%" is too strong.

We agree with the reviewer. This statement has now been revised to adopt a more measured tone (Discussion, Page 22, Paragraph 3).

4) As mentioned in the summary of my review, it is, in my opinion, necessary to highlight the fact that HVE was present despite meticulous matching. This is actually the most important finding of this study – the failure to remove HVE through matching is crucial information for the whole bulk of observational studies on COVID-19 vaccination effectiveness. I suggest adding something along the lines that "Despite meticulous matching, large healthy vaccinee effect (aHR for non-COVID death in vaccinated vs unvaccinated of 0.35) was observed in the first six months of follow-up", probably to the Discussion as well as to the Abstract.

We fully agree with the reviewer, and this has now been indicated in both the Abstract and the Discussion as suggested (Abstract and Discussion, Page 20, Paragraph 5)

5) Seeing the virtually zero effect of booster vaccination, I would be very much interested in seeing the comparison of two doses vs the unvaccinated for the "after-booster" period (not saying that this is a must, I understand it is a lot of work; it could come as a new paper).

First of all, thank you for the thoughtful comment and insightful suggestion. We agree with the reviewer that this is a topic of great interest and forms part of our other ongoing work examining the long-term effects of COVID-19 vaccines. However, given the extensive length of this manuscript (about 6000 words) and the density of results and analyses already presented, we feel it is more appropriate to reserve this analysis for a separate study specifically focused on vaccine effectiveness, rather than the healthy vaccinee effect.

6) Calculating and reporting HVE over a period of several years is not optimal. The authors discuss this very well but still analyze the combined HVE+indication bias over the whole period and report it as one of the most prominent findings. I think that the individual periods (in which these two biases can be to a large degree separated) are much more interesting and important. It would be very helpful to calculate and compare vaccine effectiveness against severe, critical, or fatal COVID for each of the periods shown in Figure 2 and present it along with the results shown in Figure 2 now. This should be ideally also done separately for the most vulnerable group of 50+ years of age.

Excellent idea, thank you. This sub analysis has now been incorporated, examining overall effectiveness as well as effectiveness within the subgroups of individuals aged <50 years and those aged ≥50 years (Figure 2—figure supplement 1). This addition has been also described in the Results section (Results, Page 19, Paragraph 1 and Page 20, Paragraph 4).

We included the results of this analysis as a separate figure because it was not feasible to perform the analysis for all the time intervals of Figure 2. This limitation arose from the relative rarity of severe forms of COVID-19. Additionally, the figure addresses a different outcome—severe COVID-19 as opposed to non-COVID-19 death. However, the analysis was conducted for the most critical time periods: 1–6 months and >6 months post vaccination (Figure 2—figure supplement 1).

7) Was there any period with testing bias in Qatar? In other words, was there any period when the vaccinated did not have to test for COVID-19 while the unvaccinated had to test? If so, could it have influenced the results of your study? I think the paper would benefit from this discussion.

Yes, there were time periods when testing requirements, such as those for travel, varied between vaccinated and unvaccinated individuals. However, we do not believe this variability would have impacted the findings of this study. This is supported by the sensitivity analysis reported in the manuscript (Table 4), as well as an additional sensitivity analysis included in our response to Reviewer 1 (please see the response above and Figure 1 in this document), where the matching requirement for a SARS-CoV-2-negative test was removed. The results of these analyses are consistent with those of the main analysis, further reinforcing the robustness of our findings.

8) I think that the peculiar composition of Qatar's population is leading to problems with low numbers, especially in the elderly population most vulnerable to COVID-19, is basically the only limitation of this study. However, the reported pattern is in line with what was observed elsewhere and the authors did, as mentioned above, an excellent job in making up for this limitation by meticulous matching so I consider their work highly relevant and well performed.

We sincerely appreciate the reviewer's thoughtful feedback and useful suggestions, which have enriched this study and its contribution to the literature.

Reviewer #3 (Recommendations for the authors):The authors have conducted cohort studies using routinely collected data from Qatar. These datasets have been the basis of a series of high-quality and influential studies that were conducted in Qatar into COVID-19 vaccine effectiveness (VE) during the pandemic.

We thank the reviewer for the time and effort put into this review, the assessment of our work, and the constructive feedback on our manuscript that enriched it and improved its readability. Please find below a point-by-point reply addressing each of the reviewer's comments.

In this paper the authors are less focused on VE and more on defining the degree and temporal profile of a possible early 'healthy vaccinee effect' followed by an 'indication bias'. The category of 'non-COVID' death is a potentially useful negative outcome control. Deviation from the null indicates bias if the vaccines truly have no effect on the study outcomes. However, its value here is reduced by the lack of information on cause of death. In this study it may be a reasonable directional bias indicator but does not specify the cause of bias. I believe their methods and analyses are robust and they are observing real effects. However, the authors do not present direct data on causes of death or variables that would help define healthy vaccinees or underpin the claim of indication bias. I think they need to provide a more comprehensive defence of their conclusions.

We agree with the reviewer that a limitation of this study is the lack of data on causes of death other than COVID-19. As a result, while the study provides an indicator of directional bias, it does not identify the specific cause of this bias. This limitation was acknowledged in the original manuscript. However, the study does provide clear evidence of the existence of the healthy vaccinee effect in rigorously conducted vaccine effectiveness studies, characterizing its effect size and temporal profile, which is the primary research question addressed in the study. This point has now been clarified and emphasized in the revised manuscript (multiple instances in the manuscript).

The manuscript has been also substantially revised and enhanced to include additional clarifications, a more detailed discussion of potential biases, and the inclusion of further analyses, all of which reinforce the study findings (changes incorporated throughout the manuscript).

1) It would be helpful if the authors defined the boundaries of the 'healthy vaccinee effect' they are investigating with their data. The closest to a definition I could find was in the Introduction where the authors state 'the healthy vaccinee effect occurs when healthier or health-conscious individuals are more likely to receive vaccination'. In this setting 'health-conscious individuals' are making a deliberate choice to be vaccinated, and this is consistent with other choices they make (exercise, avoiding smoking and excess alcohol, eating a healthy diet, undergoing screening). At times in the paper the authors appear to equate the healthy vaccinee effect with the absence of underlying disease and, at its most extreme, the absence of disease that would lead to avoidance of vaccination because of clinical futility. There is no doubt that omission of vaccination is practiced in patients with terminal disease, but I think it should be regarded as a separate bias and its absence does not denote a 'healthy vaccinee effect'. I think this needs clarification.

This study builds on research regarding this effect within the context of influenza vaccine effectiveness studies, as outlined in the Introduction (Introduction, Page 5, Paragraph 3 and Page 6, Paragraphs 1-2). To maintain consistency, we have adhered to similar definitions and conventions. While we appreciate the reviewer's suggestion, we believe it is important to align with established definitions and conventions to avoid causing confusion in this area of research. However, we have now, as suggested, explicitly defined both the healthy vaccinee effect and the indication effect in the manuscript (Methods, Page 9, Paragraph 4 and Page 10, Paragraph 1).

2) I think the overall relationship between comorbidity and frailty and vaccination choice needs deeper discussion. The authors cite community studies of influenza vaccination, which generally have lower population coverage than COVD-19 vaccination, allowing more room for selection biases. While older independent community members who are active may demonstrate a greater preference for influenza vaccination, this will not be the case for those in long-term care facilities, where a common policy is to vaccinate all residents.

We agree with the reviewer on the importance of further understanding the relationship between comorbidity and frailty and vaccination behavior. To address this, we have added a paragraph discussing this point and emphasizing the need for collecting direct primary data to better elucidate this relationship (Discussion, Page 23, Paragraph 4). Additionally, we highlighted the importance of examining how this relationship may differ across various vaccine types (Discussion, Page 23, Paragraph 4).

3) The authors also mention confounding by indication, which with COVID-19 has been influenced by government policies to provide early vaccination to high-risk groups and healthcare workers. As they say, this runs counter to the presumed healthy vaccinee effect. These policies change quickly and can introduce time-varying exposures in certain groups. I am uncertain if vaccination policies were considered as potential sources of bias – but they may be controlled by the decision to match on calendar week. The same comment applies to 'environmental risk' – variation in community attack rates over time and region.

Excellent point, thank you. As the reviewer has pointed out, vaccination policies evolve over time, potentially introducing time-varying exposures in different groups. However, as noted by the reviewer, we matched participants by calendar week and followed them starting from this time point, ensuring that each matched pair was recruited into the cohorts during a period with similar vaccination policies and practices. This approach also accounts for variations in community attack rates over time. This point has now been clarified in the revised manuscript (Methods, Page 12, Paragraph 3).

Moreover, vaccination policy in Qatar was primarily based on factors already matched in this study, such as age and coexisting conditions. To clarify this point and vaccination rollout, a subsection has been added to describe the rollout strategy employed in Qatar, including the initial target groups prioritized for vaccination (Methods, Page 8, Paragraph 3 and Page 9, Paragraphs 1-2). Additionally, a figure has been included to illustrate the rollout of both the primary vaccination series and booster doses (Figure 1—figure supplement 1).

4) The authors seem somewhat dismissive of misclassification of deaths as contributing to their findings. As noted, they don't have comprehensive cause of death data. I agree that a non-specific immunostimulant benefit of the vaccines is unlikely. However, some COVID-related deaths may be missed during coding. For instance, someone who dies of a heart attack or stroke 4 weeks after a serious COVID illness. The coding decision could also be influenced by knowledge of vaccination status as it appears that coders were unblinded.

We apologize for the confusion regarding this point. The classification of COVID-19 deaths followed a detailed and rigorous process, using strict criteria rather than a simplistic review of patient charts or reliance solely on ICD-10 codes. As a result, it is unlikely that misclassification occurred at a level that could meaningfully affect the study findings, particularly given that COVID-19 deaths in Qatar were much lower than overall mortality rates (Discussion, Page 24, Paragraph 2). Additionally, we refer the reviewer to our earlier detailed studies on COVID-19 mortality for further context [3-5].

For example, the specific scenario the reviewer highlights – a heart attack or stroke occurring four weeks after a severe COVID-19 illness – would not have been overlooked as long as the case met the defined criteria. According to the case definition, a death is classified as a COVID-19 death if there was no period of *complete recovery* from COVID-19 between the illness and death.

To address this matter, we have now expanded the description of this aspect of the methods into a subsection, detailing the classification process for COVID-19 deaths (Methods, Page 10, Paragraphs 3-4 and Page 11, Paragraphs 1-3).

5) But misclassification of outcomes is not the only potential source of bias. Because the authors don't have direct evidence to support healthy user bias and confounding by indication, I think they should use a bias framework (e.g., ROBINS-I) to discuss and (as appropriate) reject the other potential causes of bias6) I am not arguing their assertions are wrong. But I think their language is over-confident and they need to make a more comprehensive case to back their conclusions.

We address these two comments together, as they are closely related. As the reviewer has noted, this study builds upon a substantial body of prior research that we have conducted on COVID-19 using similar study designs and the same national databases. Over the course of five years conducting these studies, we have engaged in an ongoing process to investigate a wide range of potential biases and limitations, factoring frameworks such as ROBINS-I and drawing on other relevant literature. This approach explains why the limitations sections in our publications, including the present study, are typically extensive—not due to lack of rigor in the studies themselves, but because of our comprehensive examination of various potential sources of bias and limitations.

Moreover, what is included in the final publication typically represents only a subset of the potential biases and limitations we assess during the study design and implementation process. In each publication, we focus on highlighting those aspects most relevant to the specific study.

Informed by ROBINS-I, other related literature, and prior literature on the investigated effects, as well as our previous work using similar study designs on these national databases, this manuscript has been substantially revised. The revisions include a more detailed discussion of potential biases and limitations, along with additional analyses, all of which reinforce the robustness of the study findings (changes incorporated throughout the manuscript). Notably, the limitations section now spans four pages, addressing various forms of potential bias and limitations and evaluating whether and how they might influence the study results (Discussion, Pages 23-26).

7) This is an accomplished group performing sophisticated studies. However, the level of self-citation seems excessive. In some cases, it is justified to anchor the current work. But there is a large literature on the effects of COVID-19 on COVID mortality with a significant number of studies that also reported all causes of non-COVID mortality. This is not reflected in the authors' choice of references.

We thank the reviewer for the positive assessment of our work and acknowledge the observation of substantial proportion of self-citations in the manuscript. This, however, reflects the complexity of the study and the necessity of thoroughly explaining the methods, particularly in a study investigating bias. The cited prior work was included to provide additional details on the methods, describe the databases in greater depth, avoid redundancy in explanations, offer the rationale for certain methodological choices, establish context for the study design, discuss potential biases and limitations, and support and justify specific arguments. It is worth noting that the vast majority of self-citations are in the Methods section or the limitations part of the Discussion. For example, the Introduction section does not include any self-citations.

We also recognize that there is a substantial body of literature on COVID-19 mortality and non-COVID mortality that is not cited in this manuscript. However, some relevant COVID-19 mortality literature was included in the original manuscript (such as [6-8]), and additional references were cited in our previous studies specifically focused on COVID-19 mortality [3-5]. This study, however, is not a general investigation of COVID-19 or all-cause mortality but a focused examination of the healthy vaccinee effect in vaccine effectiveness studies. Other COVID-19 mortality literature was cited when directly relevant, such as in discussion on documented versus undocumented COVID-19 deaths (Discussion, Page 24, Paragraph 2).

To address the reviewer's comment, we have carefully reviewed the manuscript to reassess the relevance of each citation and have added other pertinent references from the global literature where appropriate (throughout the manuscript).